# Neural Fourier Transform
# for Multiple Time Series Prediction

**Noam Koren**                                                   *noam.koren@campus.technion.ac.il*
*Computer Science*
*Technion - Israel Institute of Technology, Haifa, Israel*

**Daniel Freedman**                                                   *dfreedman@tauex.tau.ac.il*
*Applied Mathematics*
*Tel Aviv University, Tel Aviv, Israel*

**Kira Radinsky**                                                   *kiraradinsky@gmail.com*
*Computer Science*
*Technion - Israel Institute of Technology, Haifa, Israel*

**Reviewed on OpenReview:** *https://openreview.net/forum?id=OGBjIwRuVp*

## Abstract

Multivariate time series forecasting is an important task in various fields such as economic planning, healthcare management, and environmental monitoring. In this work, we present a novel methodology for improving multivariate forecasting, particularly, in data sets with strong seasonality. We frame the forecasting task as a Multi-Dimensional Fourier Transform (MFT) problem and propose the Neural Fourier Transform (NFT) that leverages a deep learning model to predict future time series values by learning the MFT coefficients. The efficacy of NFT is empirically validated on 7 diverse datasets, demonstrating improvements over multiple forecasting horizons and lookbacks, thereby establishing new state-of-the-art results. Our contributions advance the field of multivariate time series forecasting by providing a model that excels in predictive accuracy. The code of this study is publicly available[1].

## 1 Introduction

Time series forecasting, the task of predicting future values based on historical observations, is a pivotal task in various fields such as economics, medicine, and environmental science. This task becomes particularly complex when dealing with *multiple time series*, also commonly referred to as *multivariate* time series, where multiple interdependent variables evolve simultaneously over time. For example, in weather forecasting, variables such as temperature, humidity, and wind speed interact; in finance, the prices of interconnected assets move together.

Recent advancements in time series forecasting have transitioned from conventional statistical approaches Lütkepohl (2005); Pfeffermann & Allon (1989) to sophisticated machine learning techniques, notably deep learning Lea et al. (2016); Nie et al. (2022); Oreshkin et al. (2019). However, the field still grapples with a scarcity of effective models for multivariate time series prediction, especially in settings where the data has pronounced seasonality. In many domains such as weather and energy, seasonal signals often span multiple variables simultaneously. These cross-variable seasonal dependencies cannot be adequately captured by univariate models or per-variable frequency analysis. Extending models from univariate to multivariate settings entails more than an increase in input dimensions; it demands a complex redesign to capture

---

[1]https://github.com/2noamk/NFT.git

the interdependencies among multiple variables effectively. We posit that a more expressive multivariate treatment of seasonality can significantly improve forecasting accuracy.

In this paper we introduce a novel architecture for multivariate predictions, called the Neural Fourier Transform (NFT). Fourier Transforms (FT), known for their simplicity and interpretability Carslaw (1921), decompose time series into frequency components, thus aiding in the identification and understanding of periodic patterns and trends. This makes them particularly well-suited for datasets with strong seasonality, where capturing and isolating periodic signals is essential. Additionally, FT provides a transparent view of the periodicities driving the data, enabling explicit representations of time series components and offering valuable insights into their underlying trends and seasonal structures.

NFT leverages Multi-Dimensional Fourier Transforms (MFT) Tolimieri et al. (2012) for the task of time series prediction. The NFT algorithm predicts the MFT coefficients of future timestamps using Temporal Convolutional Network (TCN) layers Lea et al. (2016), and then reconstructs the future values via an inverse MFT. To the best of our knowledge, this is the first application of MFT in the context of multivariate time series forecasting. NFT creates an explicit representation of seasonality and trend components, making it especially effective on datasets with strong periodic patterns, where modeling these components is essential for high-accuracy forecasting.

Specifically, NFT utilizes a 2-Dimensional Discrete FT (2D-DFT), implemented via two separate 1D-DFT operations: one along the time axis and another across the variables. This dual application enables NFT to capture critical patterns not only in the temporal evolution of each variable but also in the relationships among variables. By transforming the time axis, NFT exposes dominant temporal frequencies per variable. The additional transformation across variables captures how these temporal frequencies vary across different series, revealing shared or synchronized seasonal dynamics. This approach provides a more holistic understanding of multivariate seasonal structure, in contrast to previous methods that address the frequency components of each time series independently. NFT thus enables a richer representation of both individual and cross-variable seasonal behaviors, which is crucial for accurate multivariate time series forecasting. This represents a significant advancement over the 1D-DFT utilized in prior research Oreshkin et al. (2019); Wu et al. (2022); Ye et al. (2024); Zhou et al. (2022b).

NFT is empirically evaluated using 7 diverse datasets. The results demonstrate consistent improvements over 17 state-of-the-art (SOTA) models across multiple prediction horizons and lookbacks, highlighting the model's robustness, and showcasing its potential to advance multivariate time series forecasting.

The contributions of this work are threefold:
1. We formulate the problem of multiple time series prediction as a Multi-Dimensional Fourier Transform (MFT). To the best of our knowledge, this is the first work to model multiple time series in this manner. This representation offers significant advantages, including enhanced ability to capture seasonal patterns for multivariate time series analysis.
2. We propose the NFT algorithm to learn the Multivariate Fourier coefficients of unknown future timesteps.
3. NFT is evaluated on 7 diverse datasets under standard settings, achieving superior performance compared to 17 SOTA methods. Its effectiveness is particularly pronounced in datasets with strong seasonality, as demonstrated through empirical experiments on both synthetic and real data.

## 2 Related Work

### 2.1 Multivariate Time Series Forecasting

Multivariate time series forecasting has been extensively studied due to its wide range of applications. Traditional methods such as Vector Autoregression (VAR) Lütkepohl (2005) and Multivariate Exponential Smoothing Pfeffermann & Allon (1989) model linear dependencies between variables. However, their performance often degrades with increasing complexity and nonlinear interactions across series.

With the rise of deep learning, models like Long Short-Term Memory (LSTM) networks Yu et al. (2019) and Gated Recurrent Units (GRUs) Dey & Salem (2017) demonstrated improved performance by capturing temporal dependencies through recurrence. Convolutional approaches such as Temporal Convolutional Net-

works (TCNs) Lea et al. (2016) are effective for multivariate forecasting, as they jointly process all variables and capture local patterns through convolutional filters. TCNs have proven competitive on several forecasting benchmarks Zhao et al. (2017). Moreover, Graph Neural Networks (GNNs) have been adopted for multivariate time series forecasting because they can model complex relational structures between variables. Models such as Graph WaveNet Wu et al. (2019), StemGNN Cao et al. (2021) and FourierGNN Yi et al. (2023b) leverage graph-based representations to capture inter-variable dependencies.

More recently, Transformer-based architectures have been adapted for time series. Autoformer Wu et al. (2021) and FEDformer Zhou et al. (2022b) introduce attention mechanisms tailored for forecasting, often incorporating frequency-domain processing. However, recent studies, including DLinear Zeng et al. (2023), suggest that Transformers are not always optimal for prediction, particularly under resource or data constraints. Although transformers such as PatchTST Nie et al. (2022), Time-LLM Jin et al. (2023), and iTransformer Liu et al. (2023) surpass Dlinear.

These insights have led models to leverage alternative structures tailored to time series data. SOFTS Han et al. (2024) aggregates information across variables into a shared core, then distributes it back to refine each forecast. CycleNet Lin et al. (2024a) and SparseTSF Lin et al. (2024b) process each variable independently using shared model parameters. U-Mixer Ma et al. (2024) integrates MLP-Mixer blocks in a U-Net, so the mixer captures patterns across variables.

While the above models focus on inter-variable dependencies in the time domain, they overlook the seasonal patterns that extend across variables. These cross-variable periodicities evolve jointly over time and require frequency-domain representations. Such representations can be naturally obtained using the Fourier Transform, which reveals underlying frequency structures that are not easily observed in the time domain.

## 2.2 Fourier Transform in Time Series Models

Frequency-domain methods have long been central to time series analysis, enabling the identification of latent periodic structures that are often difficult to observe in the time domain Nussbaumer & Nussbaumer (1982); Yi et al. (2023a). The integration of these techniques into neural forecasting models has emerged as a promising direction, offering both performance gains and enhanced interpretability, especially in settings where the data has strong seasonality.

A prominent example is the N-BEATS algorithm Oreshkin et al. (2019), designed for *univariate* forecasting. N-BEATS decomposes time series into interpretable components using distinct architectural blocks: a Trend Block employing polynomial basis functions, and a Seasonality Block based on the Discrete Fourier Transform (DFT). While effective in its domain, N-BEATS is limited to univariate forecasting. Extending it to multivariate forecasting requires more than increasing input dimensions; it demands capturing the complex interdependencies among multiple time series.

In multivariate forecasting, several recent models integrate Fourier-based operations to improve temporal representation. Autoformer Wu et al. (2021), FEDformer Zhou et al. (2022b), FourierGNN Yi et al. (2023b), and ATFNet Ye et al. (2024) use Fourier Transforms to model long-term or global dependencies. TimesNet Wu et al. (2022), TSLANet Eldele et al. (2024), and TimeMixer Wang et al. (2024) extract multi-scale periodic patterns. FiLM Zhou et al. (2022a) and FRNet Zhang et al. (2024) apply frequency-aware modules to capture complex dynamics.

However, these models apply the Fourier Transform to each variable independently, treating them as separate univariate signals. This approach overlooks cross-variable frequency relationships, which are often crucial for accurate multivariate time series forecasting. Capturing such dependencies requires richer spectral representations that can account for both individual and inter-variable frequency dynamics.

## 2.3 Multidimensional Fourier Transform (MFT)

While the standard DFT operates over a single dimension (typically time), the *Multidimensional Fourier Transform (MFT)* Tolimieri et al. (2012) generalizes it to multi-dimensional data. This extension has found success in domains such as image analysis Rao et al. (2021); M. S. Rzeszotarski (1983); Soon & Koh (2003),

but its use in multivariate time series remains largely unexplored. Applying MFT to multivariate forecasting opens new avenues for learning patterns not only across time but also across variables, potentially uncovering joint periodicities and structured dependencies that 1D frequency models may miss.

This study proposes a novel approach based on the MFT, comparing it with a broad range of state-of-the-art methods including TimeMixer, U-Mixer, SparseTSF, SOFTS, CycleNet, FourierGNN, ATFNet, FRNet, TSLANet, FiLM, iTransformer, TimesNet, PatchTST, DLinear, TCN, N-BEATS, and VAR.

## 3   Problem Statement

In this study, we address the multivariate point forecasting problem in discrete time. We are given a series history of length $T$, $\mathbf{y} = [y_1, \ldots, y_T] \in \mathbb{R}^{M \times T}$, and our task is to predict the matrix of future values $\mathbf{Y} = [y_{T+1}, \ldots, y_{T+H}] \in \mathbb{R}^{M \times H}$, where $y_t \in \mathbb{R}^M$ for each $t = 1, \ldots, T + H$, $M$ is the number of variables, and $H$ is the forecast horizon.

To simplify, we consider a lookback window of length $t \leq T$, ending at the most recent observation $y_T$. This window serves as the input to our model and is denoted by $\mathbf{X} \in \mathbb{R}^{M \times t} = [y_{T-t+1}, \ldots, y_T]$. The forecast of $\mathbf{Y}$ is represented as $\hat{\mathbf{Y}}$.

## 4   Neural Fourier Transform (NFT)

The Neural Fourier Transform (NFT) is a novel forecasting architecture designed to capture seasonal and trend components in multivariate time series. Its core mechanism is based on a Multi-Dimensional Discrete Fourier Transform (MFT), specifically implemented using a 2D Discrete Fourier Transform (2D-DFT). This approach decomposes the data into fundamental frequency components, across both the time axis and the variable axis, enabling the identification of complex seasonal patterns that may arise from interactions between variables. Such patterns are often missed when analyzing each series independently. Figure 1 illustrates the overall structure of the NFT model.

### 4.1   Multivariate 2D-DFT Representation

The 2D-DFT of the target forecast matrix $\mathbf{Y} \in \mathbb{R}^{M \times H}$ is computed by applying two sequential 1D-DFT operations: the first along the temporal axis and the second across the variable axis Tolimieri et al. (2012), which makes 2D-DFT not a fundamentally different operation, but the composition of two 1D-DFTs applied along orthogonal axes.

First, a 1D-DFT is applied column-wise using the Fourier matrix $\mathbf{F}_M$, producing an intermediate matrix $\mathbf{Z}$:

$$\mathbf{Z} = \mathbf{F}_M \mathbf{Y}. \tag{1}$$

Next, a second 1D-DFT is applied row-wise with the Fourier matrix $\mathbf{F}_H$, yielding the matrix of Fourier coefficients $\mathbf{C}$:

$$\mathbf{C} = \mathbf{Z} \mathbf{F}_H^\top = \mathbf{F}_M \mathbf{Y} \mathbf{F}_H^\top. \tag{2}$$

The inverse transform reconstructs the time-domain forecast:

$$\hat{\mathbf{Y}} = \mathbf{F}_M^\top \mathbf{C} \mathbf{F}_H. \tag{3}$$

NFT's objective is to learn the coefficient matrix $\mathbf{C}$ corresponding to the unknown future values of $\mathbf{Y}$. Once estimated, these coefficients are mapped back through the inverse transform in Equation 3 to produce the forecast.

It is important to note that applying a Fourier Transform across the variable dimension is unconventional. Unlike the temporal axis, features do not possess a natural sequential order, and thus treating them as if they were ordered may appear arbitrary. The motivation for doing so is to uncover cross-variable seasonal

structures that are otherwise missed by per-variable transforms. As shown in Section 6.3, such cross-variable frequency patterns indeed arise in real-world datasets and are naturally captured by the 2D-DFT. This choice introduces a potential sensitivity to feature ordering, which we systematically analyze in Section 6.4, where we demonstrate that NFT remains robust under variable permutations.

## 4.2 Fourier Matrix Construction

Unlike traditional Fourier matrices, NFT introduces additional flexibility by customizing $\mathbf{F}_H$ to support varying frequency resolutions. The matrix $\mathbf{F}_M \in \mathbb{R}^{M \times M}$ is square and corresponds to the number of variables in the dataset. In contrast, $\mathbf{F}_H \in \mathbb{R}^{N \times H}$ is generally non-square, where $N$ denotes the Fourier order and $H$ is the forecast horizon. This design enables NFT to adapt the spectral granularity of the transformation to the specific seasonal characteristics of the data.

$$
F_M = \begin{bmatrix} \cos(2\pi \cdot 0 \cdot \frac{0}{M}) & \cdots & \cos(2\pi \cdot 0 \cdot \frac{M-1}{M}) \\ \vdots & \ddots & \vdots \\ \cos(2\pi \cdot \frac{M}{2} \cdot \frac{0}{M}) & \cdots & \cos(2\pi \cdot \frac{M}{2} \cdot \frac{M-1}{M}) \\ \sin(2\pi \cdot 0 \cdot \frac{0}{M}) & \cdots & \sin(2\pi \cdot 0 \cdot \frac{M-1}{M}) \\ \vdots & \ddots & \vdots \\ \sin(2\pi \cdot \frac{M}{2} \cdot \frac{0}{M}) & \cdots & \sin(2\pi \cdot \frac{M}{2} \cdot \frac{M-1}{M}) \end{bmatrix}, F_H = \begin{bmatrix} \cos(2\pi \cdot 0 \cdot \frac{0}{H}) & \cdots & \cos(2\pi \cdot 0 \cdot \frac{H-1}{H}) \\ \vdots & \ddots & \vdots \\ \cos(2\pi \cdot \frac{N}{2} \cdot \frac{0}{H}) & \cdots & \cos(2\pi \cdot \frac{N}{2} \cdot \frac{H-1}{H}) \\ \sin(2\pi \cdot 0 \cdot \frac{0}{H}) & \cdots & \sin(2\pi \cdot 0 \cdot \frac{H-1}{H}) \\ \vdots & \ddots & \vdots \\ \sin(2\pi \cdot \frac{N}{2} \cdot \frac{0}{H}) & \cdots & \sin(2\pi \cdot \frac{N}{2} \cdot \frac{H-1}{H}) \end{bmatrix}
$$

These matrices enable the model to encode both variable-wise and temporal frequency patterns in a structured and interpretable form.

## 4.3 Learning Fourier Coefficients with TCNs

To estimate the Fourier coefficients $\mathbf{C}$ from past observations $\mathbf{X} \in \mathbb{R}^{M \times T}$, NFT employs Temporal Convolutional Networks (TCNs) Zhao et al. (2017). TCNs preserve the multidimensional structure of the input, enabling them to capture both local and long-range dependencies across variables without the need to flatten the data.

While one might consider using traditional Fully Connected (FC) layers, as done in models like N-BEATS, such an approach requires flattening the input, which can lead to the loss of inter-variable relationships and increased computational overhead. In contrast, TCNs operate directly on inputs of shape ($batch\ size \times M \times T$), making them particularly well-suited for multivariate coefficient learning. Moreover, we did not adopt attention-based architectures such as Transformers, since their quadratic complexity in sequence length makes them less efficient for the long horizons considered in our experiments. (Sec. 6.6 confirm this design choice: replacing TCN layers with LSTM, SSM, or FC layers consistently degraded performance across all datasets.)

## 4.4 Modeling Non-Seasonal Components: The Trend Block

In addition to capturing seasonality, NFT includes a polynomial Trend Block to model non-periodic patterns. Each variable's future trajectory is approximated using a low-degree polynomial:

$$
\hat{\mathbf{Y}} = \mathbf{A}\mathbf{P} \tag{4}
$$

where $\mathbf{A} \in \mathbb{R}^{M \times d}$ contains the polynomial coefficients, and $\mathbf{P} \in \mathbb{R}^{d \times H}$ is a Vandermonde matrix constructed from a normalized time vector.

The Trend Block also employs TCN layers to estimate $\mathbf{A}$, enabling tailored trend modeling per variable.

### 4.5 Block-Wise Stacking

NFT adopts a residual block stacking strategy inspired by N-BEATS. It utilizes separate stacks of Seasonality and Trend Blocks. Each block receives the residual from the previous block and produces two outputs: a backcast and a forecast.

To produce these two outputs, each block learns two sets of coefficients: $C_{\text{back}}$, which reconstructs the observed past window, and $C_{\text{fore}}$, which parameterizes the forecasted future window. These matrices serve distinct roles within the residual architecture: $C_{\text{back}}$ captures the structure already present in the input, while $C_{\text{fore}}$ captures the structure of the unknown future. In Seasonality Blocks, the coefficients are mapped to the time domain using the inverse 2D-DFT, whereas in Trend Blocks they are mapped using the polynomial trend basis. This yields each block's backcast and forecast components.

The backcast is the block's attempt to reconstruct the portion of the input signal it is responsible for modeling. By subtracting this backcast from the running residual, each subsequent block focuses only on the unexplained structure. This recursive refinement progressively decomposes the time series into distinct, interpretable components, while the forecasts from all blocks are summed to produce the final prediction.

Let $\mathbf{X}$ denote the input time series. The first stack consists of $N$ Seasonality Blocks, denoted $\mathbf{B}_s^{(i)}$ for $i = 1, \ldots, N$. The computation proceeds as follows:

$$\mathbf{R}_s^{(0)} = \mathbf{X}, \tag{5}$$

$$\mathbf{O}_{s,\text{backcast}}^{(i)}, \ \mathbf{O}_{s,\text{forecast}}^{(i)} = \mathbf{B}_s^{(i)}(\mathbf{R}_s^{(i-1)}), \tag{6}$$

$$\mathbf{R}_s^{(i)} = \mathbf{R}_s^{(i-1)} - \mathbf{O}_{s,\text{backcast}}^{(i)}. \tag{7}$$

The total forecast of the Seasonality stack is

$$\mathbf{O}_s = \sum_{i=1}^{N} \mathbf{O}_{s,\text{forecast}}^{(i)}. \tag{8}$$

The final residual $\mathbf{R}_s^{(N)}$ is passed to the Trend stack, which follows the same block-wise formulation. The total trend forecast is denoted $\mathbf{O}_t$. The final output of the NFT model is the sum of the seasonality and trend predictions:

$$\mathbf{Y}_{\text{pred}} = \mathbf{O}_s + \mathbf{O}_t. \tag{9}$$

This stacked architecture enables NFT to progressively decompose the time series into interpretable components, improving its ability to model complex temporal dynamics and multivariate dependencies.

## 5 Empirical Evaluation

### 5.1 Experimental Methodology

Our experiments focus on predicting future time steps of multivariate time series. The forecasting performance is evaluated using the mean squared error (MSE) metric, which is the standard evaluation criterion in the time-series forecasting literature Das et al. (2004); Makridakis & Hibon (2000).

$$\text{MSE} = \frac{1}{H} \frac{1}{M} \sum_{i=1}^{H} \sum_{j=1}^{M} (y_{j,T+i} - \hat{y}_{j,T+i})^2.$$

### 5.2 Datasets

We evaluate NFT on 7 benchmark datasets that are widely used in comparative studies of multivariate time series forecasting. These include Electricity, Traffic, Exchange Rate, Weather, ETTm1, 12-lead ECG, and

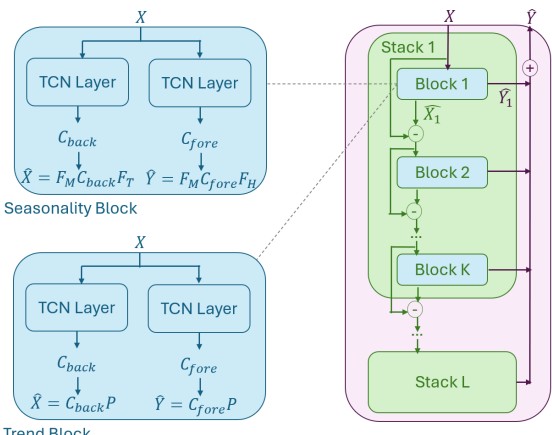

Figure 1: Architecture of the Neural Fourier Transform (NFT) model. The diagram illustrates the arrangement of Seasonality and Trend Blocks, and the use of 2D-DFT and TCNs.

36-lead EEG. All datasets were preprocessed following the configurations established in prior work Nie et al. (2022); Wu et al. (2022; 2021); Zeng et al. (2023); Zhou et al. (2022b); Raghunath et al. (2020); Batista et al. (2024); Kotecha (2018). Details on the datasets are provided in Appendix 8.1.

The ECG and EEG datasets contain collections of multivariate time series, where each series corresponds to an individual subject. For example, the ECG dataset consists of recordings from multiple patients, each represented as a 12-variable time series. In our experiments, we selected four patients from the ECG dataset and two from the EEG dataset, reporting results averaged across these individuals.

**Horizon Selection:** To obtain dataset-specific yet broadly applicable forecasting horizons, we first estimate the dominant cycle length in each dataset using standard spectral-analysis techniques—the established method for identifying the strongest repeating component in a signal Oppenheim et al. (1997); Wilks (2011). Following this methodology, we identify the dominant frequency $f_{\text{peak}}$ corresponding to the highest spectral magnitude via the Fast Fourier Transform (FFT). As is standard in signal processing, the cycle length is then computed as the reciprocal of this frequency:

$$\text{cycle length} = \frac{1}{f_{\text{peak}}}.$$

We average these cycle lengths across all variables to obtain a single, robust reference period for each dataset. We select horizons around the estimated cycle length based on this data-driven reference. A cycle-anchored, multi-horizon design of this kind has become standard in recent forecasting benchmarks, e.g., ETT Zhou et al. (2021) and WeatherBench Rasp et al. (2020), so adopting it aligns our study with prevailing practice and facilitates fair comparison with prior work. Table 1 lists the dataset statistics, estimated cycle lengths, and the horizons used in our experiments.

Table 1: Datasets Statistics

| Dataset | Exchange Rate | Weather | Traffic | Electricity |
|---|---|---|---|---|
| Variables | 8 | 3 | 862 | 321 |
| Timesteps | 7588 | 17235 | 17544 | 6304 |
| Cycle Length | 87.50 | 70.14 | 24.00 | 23.95 |
| Lookback | 96 | 96 | 96 | 96 |
| Horizons | 96, 192, 336, 720 | 96, 192, 336, 720 | 1, 16, 32, 48 | 1, 16, 32 |

| Dataset | ETTm1 | ECG | EEG |
|---|---|---|---|
| Variables | 7 | 12 | 36 |
| Timesteps | 57792 | 3989 | 176395 |
| Cycle Length | 89.12 | 69.95 | 36.60 |
| Lookback | 96 | 200 | 100 |
| Horizons | 96, 192, 336, 720 | 10, 25, 50, 100 | 1, 10, 25, 50 |

### 5.3 Baseline Models

The Neural Fourier Transform (NFT) is compared to 17 SOTA forecasting architectures, including TimeMixer Wang et al. (2024), U-Mixer Ma et al. (2024), SparseTSF Lin et al. (2024b), SOFTS Han et al. (2024), CycleNet Lin et al. (2024a), FRNet Zhang et al. (2024), TSLANet Eldele et al. (2024), FourierGNN Yi et al. (2023b), ATFNet Ye et al. (2024), iTransformer Liu et al. (2023), FiLM Zhou et al. (2022a), TimesNet Wu et al. (2022), PatchTST Nie et al. (2022), DLinear Zeng et al. (2023), Temporal Convolutional Network (TCN) Lea et al. (2016), N-BEATS Oreshkin et al. (2019) and VAR Lütkepohl (2005). Notably, FRNet, TSLANet, ATFNet, FourierGNN, TimeMixer, FiLM, TimesNet, and N-BEATS are based on 1D-DFT. Each model was tested on the 7 diverse datasets, across different forecasting horizons, to ensure a comprehensive and robust evaluation.

Each baseline model was trained strictly following its official codebase and recommended hyperparameters. The configurations for each baseline are summarized in Table 7 in Appendix 8.2.

The experiments were conducted using an NVIDIA L40 49GB GPU. We employed a batch size of 32, limited the training to 10 epochs, and used MSE as the loss function, following the methodology outlined in Wu et al. (2022). All datasets were partitioned using the standard 7:1:2 train/validation/test split. Since this is temporal data, we employ strictly temporal (non-shuffled) splits to prevent data leakage, ensuring no overlap between segments and no shuffling across time. Specifically, if the data consists of T time steps, we divide it into three segments: the first 0.7T for training, the next 0.1T for validation, and the final 0.2T for testing.

Hyperparameter tuning was performed on the validation split to ensure optimal model performance.
The exact hyperparameters used, along with the details of the hyperparameter tuning process, are provided in Appendix 8.2. This includes configurations for the NFT model, variations in Fourier granularity and polynomial degrees, and dataset-specific stack configurations, ensuring transparency and reproducibility of the experiments.

## 6 Empirical Results

### 6.1 Main Result

The results, displayed in Table 2, highlight the NFT model's strong performance across a diverse range of datasets. Accuracy was measured via the mean squared error (MSE) metric, which is widely recognized in forecasting literature Das et al. (2004); Hyndman & Koehler (2006); Makridakis & Hibon (2000). The model achieves SOTA performance and significant MSE reductions, with average improvement percentages of **20.05%** in ECG, **10.46%** in Electricity, **5.54%** in EEG, **4.40%** in Traffic, **3.43%** in ETTm1, **1.92%** in Weather, and **1.16%** in Exchange.
The improvement was calculated by comparing the NFT model to the best-performing baseline per each horizon. For each dataset we averaged the MSE improvement percentages across time horizons.
To ensure fair and consistent comparisons, we used the reported results of the respective articles when available, otherwise, we ran the author's code.

**Significance of Results.** To evaluate the statistical robustness of NFT, we conducted each experiment 20 times with different random seeds and computed the mean, standard deviation (SD), and 95% confidence intervals (CI) for the MSE metric. In Table 2, results in which the *upper bound* of NFT's confidence interval remains below that of the second-best model are marked with an asterisk (*), indicating **statistically significant improvement**. Full numerical results, are provided in Table 3.

Additionally, we evaluated the performance of NFT using the mean absolute error (MAE) metric Willmott & Matsuura (2005). The results, presented in Table 9 in Appendix 8.3, exhibit a similar trend to what was observed with the MSE metric, highlighting NFT's strong performance, with average improvement percentages of **30.26%** in EEG, **3.75%** in Exchange, and **2.15%** in Weather. As with MSE, the improvements were calculated relative to the best-performing baseline per horizon, with results averaged across time horizons.

Table 2: MSE results across models and datasets. Best results are in bold and the second-best are underlined. Results marked with an asterisk (*) indicate cases where NFT's upper Confidence Bound remains below that of the second-best model.

| Data | Horizon | NFT | TimeMixer | U-Mixer | SparseTSF | SOFTS | CycleNet | FourierGNN | ATFNet | FRNet | TSLANet | iTransformer | FiLM | TimesNet | PatchTST | DLinear | TCN | N-BEATS | VAR |
|---|---|---|---|---|---|---|---|---|---|---|---|---|---|---|---|---|---|---|---|
| Exchange | 96 | **0.060** | 0.108 | 0.087 | 0.083 | 0.136 | 0.105 | 0.215 | 0.107 | _0.081_ | 0.083 | 0.086 | 0.110 | 0.107 | 0.107 | 0.088 | 0.310 | 0.170 | 1.214 |
| | 192 | **0.164** | 0.219 | _0.171_ | 0.173 | 0.259 | 0.212 | 0.319 | 0.291 | 0.178 | 0.177 | 0.177 | 0.850 | 0.226 | 0.226 | 0.176 | 0.170 | 0.190 | 1.133 |
| | 336 | 0.422 | 0.825 | 0.352 | _0.317_ | 0.515 | 0.415 | 0.598 | 0.393 | 0.343 | 0.331 | 0.331 | 0.327 | 0.367 | 0.367 | **0.313** | 0.950 | 0.430 | 2.635 |
| | 720 | _0.670_ | 1.089 | **0.611** | 0.814 | 1.184 | 1.132 | 0.974 | 0.671 | 0.893 | 0.888 | 0.847 | 0.811 | 0.964 | 0.964 | 0.839 | 1.140 | 0.850 | 2.768 |
| Traffic | 1 | **0.200** | 0.215 | 0.257 | 0.458 | _0.213_ | 0.215 | 0.516 | 0.217 | 0.410 | 0.218 | 0.289 | 1.372 | 0.530 | 0.260 | 0.400 | 0.300 | 0.390 | 3.704 |
| | 16 | **0.320** | 0.405 | 0.439 | 0.619 | _0.328_ | 0.405 | 0.701 | 0.381 | 0.470 | 0.397 | 0.622 | 0.612 | 0.580 | 0.580 | 0.630 | 0.500 | 0.510 | 0.801 |
| | 32 | **0.390*** | 0.429 | 0.462 | 0.670 | _0.408_ | 0.429 | 0.764 | 0.418 | 0.521 | 0.431 | 0.726 | 0.640 | 0.620 | 0.630 | 0.670 | 0.490 | 0.590 | 4.110 |
| | 48 | **0.400*** | 0.453 | 0.478 | 0.719 | _0.416_ | 0.450 | 0.845 | 0.430 | 0.570 | 0.450 | 0.785 | 0.721 | 0.620 | 0.660 | 0.720 | 0.510 | 0.710 | 2.410 |
| ETTm1 | 48 | **0.270*** | 0.352 | 0.368 | 0.348 | 0.402 | 0.354 | 0.507 | 0.350 | 0.345 | 0.330 | 0.290 | 0.330 | 0.293 | _0.280_ | 0.310 | 0.430 | 0.450 | 2.085 |
| | 96 | **0.241*** | 0.320 | 0.422 | 0.365 | 0.451 | 0.391 | 0.524 | 0.327 | _0.287_ | 0.289 | 0.334 | 0.302 | 0.340 | 0.293 | 0.299 | 0.500 | 0.350 | 2.176 |
| | 192 | **0.320** | 0.361 | 0.446 | 0.441 | 0.537 | 0.442 | 0.624 | 0.370 | _0.327_ | 0.328 | 0.377 | 0.338 | 0.370 | 0.330 | 0.340 | 0.580 | 0.500 | 1.918 |
| | 336 | 0.400 | 0.390 | 0.516 | 0.508 | 0.624 | 0.506 | 0.639 | 0.402 | _0.362_ | **0.355** | 0.426 | 0.373 | 0.410 | 0.370 | 0.370 | 0.590 | 0.580 | 1.513 |
| Weather | 96 | **0.364** | 0.573 | 0.390 | 0.743 | 0.554 | 0.748 | 0.799 | 0.497 | 0.428 | 0.659 | 1.011 | 1.167 | 0.667 | 0.873 | 0.388 | 0.383 | _0.372_ | 1.001 |
| | 192 | **0.383** | 0.990 | 0.436 | 1.079 | 0.749 | 1.053 | 0.823 | 0.826 | 0.456 | 0.933 | 1.407 | 1.427 | 0.948 | 1.258 | 0.435 | 0.440 | _0.395_ | 1.285 |
| | 336 | **0.384*** | 0.696 | 0.414 | 0.829 | 0.614 | 0.828 | 1.005 | 0.682 | 0.607 | 0.757 | 1.120 | 1.004 | 0.709 | 0.962 | 0.415 | 0.420 | _0.392_ | 1.155 |
| | 720 | 0.481 | 0.747 | 0.528 | 0.807 | 0.607 | 0.843 | 1.111 | 0.717 | 0.726 | 0.767 | 1.112 | 1.015 | 0.772 | 0.940 | **0.481** | 0.521 | 0.493 | 1.124 |
| ECG | 10 | **0.172*** | 0.234 | _0.223_ | 0.286 | 0.224 | 0.243 | 0.701 | 0.247 | 0.277 | 0.290 | 0.512 | 0.842 | 0.490 | 0.570 | 0.460 | 0.420 | 0.290 | 1.424 |
| | 25 | **0.258*** | 0.640 | _0.338_ | 0.544 | 0.342 | 0.398 | 0.778 | 0.395 | 0.547 | 0.380 | 0.742 | 1.057 | 0.625 | 0.770 | 0.580 | 0.560 | 0.390 | 1.948 |
| | 50 | **0.400*** | 0.489 | 0.343 | 0.752 | 0.485 | 0.527 | 0.952 | 0.507 | 0.755 | _0.410_ | 0.987 | 1.272 | 1.160 | 0.940 | 0.660 | 0.700 | 0.490 | 1.378 |
| | 100 | **0.393*** | 0.769 | 0.500 | 0.864 | 0.601 | 0.641 | 1.023 | 0.597 | 0.835 | _0.460_ | 1.000 | 1.100 | 1.090 | 0.980 | 0.690 | 0.810 | 0.590 | 0.974 |
| EEG | 1 | **0.075*** | 0.101 | 0.090 | 0.137 | _0.084_ | 0.103 | 0.578 | 0.093 | 0.150 | 0.180 | 0.130 | 0.594 | 0.160 | 0.150 | 0.135 | 0.135 | 0.110 | 1.312 |
| | 10 | **0.159** | 0.324 | 0.244 | 0.349 | _0.163_ | 0.307 | 0.596 | 0.328 | 0.346 | 0.345 | 0.365 | 0.608 | 0.350 | 0.345 | 0.365 | 0.420 | 0.320 | 1.044 |
| | 25 | **0.238*** | 0.506 | 0.318 | 0.427 | _0.243_ | 0.393 | 0.611 | 0.406 | 0.462 | 0.410 | 0.435 | 0.624 | 0.440 | 0.420 | 0.435 | 0.430 | 0.430 | 1.373 |
| | 50 | **0.297*** | 0.686 | 0.457 | 0.496 | _0.317_ | 0.423 | 0.681 | 0.473 | 0.568 | 0.474 | 0.521 | 0.694 | 0.557 | 0.489 | 0.501 | 0.460 | 0.524 | 1.674 |
| Electricity | 1 | **0.051*** | 0.107 | 0.084 | 0.158 | 0.057 | 0.054 | 0.208 | 0.065 | 0.116 | _0.054_ | 0.070 | 0.827 | 0.140 | 0.090 | 0.070 | 0.080 | 0.060 | 2.220 |
| | 16 | **0.080*** | 0.109 | 0.148 | 0.155 | _0.097_ | 0.099 | 0.401 | 0.131 | 0.128 | _0.100_ | 0.167 | 0.181 | 0.160 | 0.200 | 0.160 | 0.120 | 0.130 | 1.771 |
| | 32 | **0.141*** | 0.148 | 0.167 | 0.180 | _0.147_ | 0.151 | 0.576 | 0.156 | 0.151 | 0.148 | 0.205 | 0.197 | 0.170 | 0.170 | 0.180 | 0.340 | 0.380 | 0.644 |

Table 3: Mean MSE results (± 95% confidence interval) over 20 runs with different random seeds.

| | Electricity | | Exchange | | Traffic | | ETTm1 |
|---|---|---|---|---|---|---|---|
| 1 | 0.051±0.001 | 96 | 0.060±0.021 | 1 | 0.200±0.015 | 48 | 0.270±0.007 |
| 16 | 0.080±0.001 | 192 | 0.164±0.019 | 16 | 0.320±0.019 | 96 | 0.241±0.008 |
| 32 | 0.141±0.001 | 336 | 0.422±0.027 | 32 | 0.390±0.011 | 192 | 0.320±0.011 |
| | | 720 | 0.670±0.059 | 48 | 0.400±0.015 | 336 | 0.400±0.012 |
| | **Weather** | | **ECG** | | **EEG** | | |
| 96 | 0.364±0.011 | 10 | 0.172±0.003 | 1 | 0.075±0.011 | | |
| 192 | 0.383±0.014 | 25 | 0.258±0.004 | 10 | 0.159±0.014 | | |
| 336 | 0.384±0.009 | 50 | 0.400±0.004 | 25 | 0.238±0.001 | | |
| 720 | 0.481±0.005 | 100 | 0.393±0.004 | 50 | 0.297±0.001 | | |

Overall, the results indicate that the NFT model provides a robust solution for multivariate time series forecasting. Its consistent performance across different datasets, horizons, and lookbacks highlights its effectiveness for interpretable multivariate time series prediction.

The baseline models against which we compared, included DFT methods (ATFNet, FourierGNN, FRNet, TSLANet, FiLM, TimesNet, TimeMixer, and N-BEATS) predominantly employing 1D-DFT approaches. The reported results illustrate that the NFT's 2D-DFT approach surpasses traditional 1D-DFT methods, providing more accurate and reliable predictions for multivariate time series. We hypothesize that the results stems from NFT model ability to capture temporal and spatial dependencies more effectively, enabling it to uncover interrelationships and patterns that 1D methods may overlook.

## 6.2 Impact of Seasonality Proportion on Performance

In this section, we analyze the performance of NFT as a function of dataset seasonality. Following the definition of Wang et al. (2006), the strength of a time series' seasonality, $F_S$, is defined as:

$$F_S = \max\left(0, 1 - \frac{\text{Var}(R)}{\text{Var}(S + R)}\right)$$

where the seasonality $(S)$, trend $(T)$, and remainder $(R)$, are calculated as described by Box et al. (2015). The strength of the trend, $F_T$, is defined analogously.

To quantify the relative contribution of the seasonality, we introduce the *Seasonal Proportion (SP)*, defined as:

$$SP = \frac{F_S}{F_S + F_T}.$$

This metric measures the relative dominance of seasonality compared to trend in the data.

### 6.2.1 Synthetic Data Experiments

To study the relationship between Seasonal Proportion (SP) and NFT performance, we generated a diverse set of synthetic time series with varying combinations of trend and seasonality strengths. These experiments are designed to evaluate model behavior in a controlled setting, where seasonality can be isolated as the primary influencing factor. This design enables a precise analysis of how forecasting performance correlates with seasonal dominance, as quantified by SP.

Time series with similar SP values were grouped, and MSE differences were averaged within each group to enable robust comparisons across models. Full details of the data generation process are provided in Appendix 8.4.

Figure 2 presents results for a 5-variable time series using a lookback window of 25 time steps and a forecast horizon of 50 time steps. As expected, when SP is below 0.46 (i.e., when trend dominates), NFT underperforms relative to other methods. However, as SP increases and seasonality becomes the dominant signal, NFT consistently achieves lower MSE, demonstrating its effectiveness in capturing seasonal patterns.

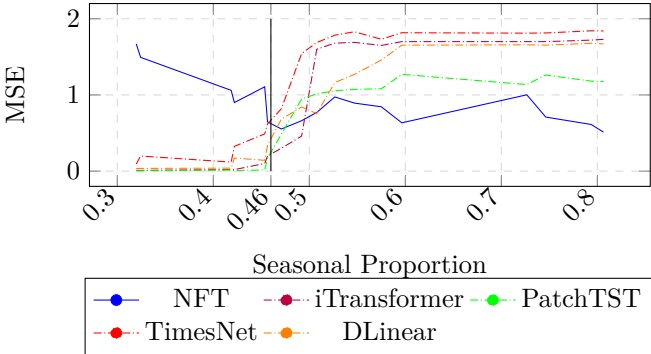

Figure 2: MSE values for NFT, iTransformer, PatchTST, TimesNet, and DLinear across varying Seasonal Proportions on synthetic data. The compared baselines represent three different modeling approaches: Transformers, 1D-DFT, and linear layers. NFT's relative performance improves as seasonality proportion increases, confirming its ability to effectively capture dominant seasonal patterns.

### 6.2.2 Real Data Experiments

While the synthetic data experiments allow controlled isolation of seasonality strength, we validate these findings on real-world datasets by examining the relationship between the SP and NFT's relative forecasting improvement.

To this end, we construct a meta-dataset in which each point represents a real-world dataset, characterized by its SP and NFT's MSE improvement over the best-performing baseline. Figure 3 visualizes these results. A polynomial trend line is fitted using least-squares regression to highlight the general trend.

The results indicate a positive correlation: datasets with higher SP values tend to exhibit greater gains. Notably, datasets such as ECG (SP = 0.51) show substantial improvements. Specifically, the slope of the fitted line is 106 when all data points are included. These findings support the hypothesis that NFT is particularly effective in settings with strong seasonal components.

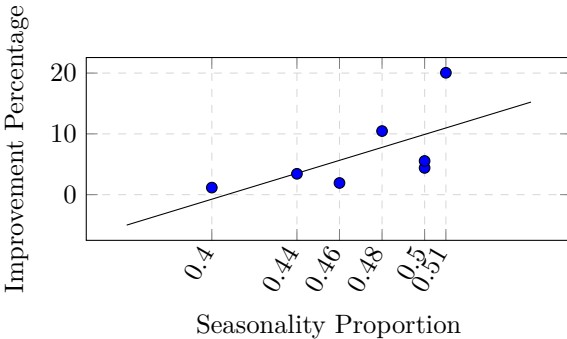

Figure 3: NFT's MSE improvement percentage on real-world datasets as a function of Seasonality Proportion (SP). Each point in the data represents a dataset (including Exchange (0.40), ETTm1 (0.44), Weather (0.46), Electricity (0.48), Traffic (0.50), EEG (0.50), and ECG (0.51)). The graph line represents a polynomial trend line fitted to the data, with an upward slope of 106 , indicating a positive correlation between SP and the NFT model's improvement percentages.

## 6.3 Visualizing Cross-Variable Seasonality with 2D-DFT

To illustrate the benefits of applying a 2D Discrete Fourier Transform (2D-DFT) in multivariate time series forecasting, Figure 4 shows the 2D frequency spectra for two representative datasets: ECG and Electricity. Each heatmap displays spectral magnitudes (on a logarithmic scale) across time frequencies (vertical axis) and variable frequencies (horizontal axis), where brighter colors indicate stronger signal energy at a given frequency pair.

These spectra expose underlying patterns of periodicity and cross-variable dynamics, which motivate NFT's frequency-aware design. Brighter regions at low time frequencies suggest smooth, regular patterns over time (e.g., cycles or trends), while brightness at high time frequencies implies noise or rapid fluctuations. Similarly, brightness concentrated at low variable frequencies indicates that many variables co-vary in a synchronized way, whereas higher variable frequencies suggest more diverse or even opposing behaviors across variables.

**ECG: Aligned Rhythmic Patterns Across Leads.** The ECG spectrum reveals high magnitude at low time and low variable frequencies, indicating clear and regular periodicity (e.g., heartbeats) that is synchronized across channels. The absence of high-frequency content along both axes suggests low noise and strong alignment among the ECG leads, where an increase in one lead tends to coincide with increases in others.

**Electricity: Shared but Desynchronized Seasonality.** The Electricity dataset exhibits strong low time-frequency components, consistent with known daily usage cycles. However, unlike ECG, the energy is distributed across a wide range of variable frequencies. This pattern reflects real-world variation: while many regions follow a daily rhythm, their peaks are often misaligned, for example, residential areas peak in the evening and commercial areas peak around noon. This "shared but desynchronized" seasonality is challenging for models that treat variables independently, but is naturally captured by NFT's joint frequency decomposition.

This analysis highlights that many real-world multivariate datasets do not simply exhibit per-variable seasonality, but instead contain structured, cross-variable frequency patterns. NFT leverages this structure by learning joint frequency-domain representations, enabling more accurate forecasting. In contrast, 1D approaches, focused solely on individual time series, struggle to capture the kind of shared but unsynchronized behavior clearly visible in the Electricity dataset. The 2D-DFT framework directly addresses this limitation by modeling correlations across both time and variable dimensions.

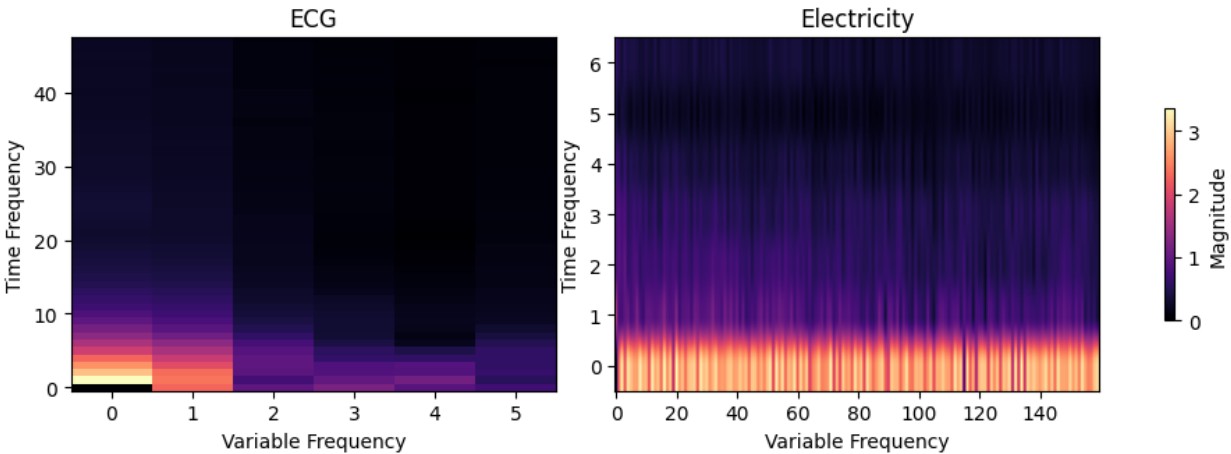

Figure 4: 2D-DFT magnitude spectra (logarithmic scale) for ECG and Electricity datasets. ECG rhythms are highly synchronized. Electricity displays shared but phase-shifted cycles across variables, as seen in the spread along the variable frequency axis.

## 6.4 Robustness to Variable Ordering

A potential concern with NFT's use of the 2D-DFT is that, unlike the temporal dimension, the variable dimension often has no canonical sequential order. Since the DFT is an order-dependent operator, reordering variables before applying the transform alters the frequency representation. This creates a conceptual challenge: how can an order-dependent operator be effective on an unordered set of variables?

To investigate this theoretical sensitivity, we visualize the spectral transformation under random permutations. As seen in Figure 5, permuting the variable order in **Electricity** does not have a drastic visual impact; the energy is already distributed across a wide range of variable frequencies in the original data. This suggests that the inherent desynchronization across nodes already populates a broad spectral range, making the representation naturally less sensitive to specific adjacency.

In **ECG**, there is more visual variability. The original data shows a concentrated spike at low variable frequencies, indicating high synchronization across leads. While permutations smear this energy across the horizontal axis (the variate-frequency dimension), the high-magnitude components at low frequencies remain constant across all permutations. This suggests that the core information regarding the rhythmic periodicities (e.g., the heartbeat) is preserved in the spectral domain regardless of variable ordering.

To quantitatively evaluate this sensitivity, we conducted a robustness analysis by randomly permuting the variable order before both training and inference. For each dataset, we performed 10 independent runs with different random permutations and computed the mean MSE and standard deviation (SD).

Table 4: NFT performance under random variable permutations averaged across all horizons (Mean MSE ± SD across 10 runs). The low standard deviation confirms that NFT is functionally robust to variable ordering.

| Dataset | NFT Shuffled Vars (MSE ± SD) |
|---|---|
| Electricity | 0.091 ± 0.008 |
| Exchange | 0.329 ± 0.015 |
| Traffic | 0.328 ± 0.003 |
| Weather | 0.403 ± 0.017 |
| ETTm1 | 0.308 ± 0.072 |
| ECG | 0.306 ± 0.089 |
| EEG | 0.192 ± 0.009 |

Figure 5: 2D-DFT magnitude spectra comparison between original variable ordering and a random permutation. In Electricity (top), the diffuse energy remains visually similar. In ECG (bottom), the high synchronization is smeared across the variate frequency axis after permutation; however, the dominant low frequency peak is preserved, explaining the model's performance stability.

The results in Table 4 demonstrate that NFT maintains stable performance despite variable permutations. We attribute this robustness to two primary factors:

1. **Global Feature Aggregation:** Unlike a Convolutional Neural Network (CNN) that relies on local adjacency (where shuffling pixels destroys local features), the DFT is a *global* operator. Every frequency bin in the variate dimension is a weighted sum of *all* variables. Thus, a permutation simply redistributes the energy across the spectral grid rather than destroying the underlying information.

2. **Neural Adaptation:** The learnable layers within the NFT act directly on the spectral coefficients. Since these layers are trained end-to-end, they learn to map the redistributed spectral energy to the target forecast. The network essentially learns the specific mapping required for a given ordering during training.

This analysis confirms that for most multivariate datasets, the 2D-DFT serves as a powerful basis transformation that exposes joint temporal-variate dynamics, which the neural backbone can exploit regardless of the initial variable indexing. Extended visualizations of these spectral shifts across nine random permutations for all datasets are provided in Appendix 8.7.

## 6.5 NFT Decomposability Results

NFT enables explainable forecasting by explicitly decomposing each time series into seasonal and trend components. It directly learns Fourier coefficients that capture periodic patterns and reconstructs the seasonal component using the inverse 2D Discrete Fourier Transform (Equation 3). The trend component is modeled separately through a polynomial approximation, where the learned coefficients reflect long-term dynamics (Equation 4).

These learned representations provide insight into the underlying temporal structure of the data. Figure 6 illustrates NFT forecasts for three variables from the Electricity dataset, along with their corresponding decomposed components.

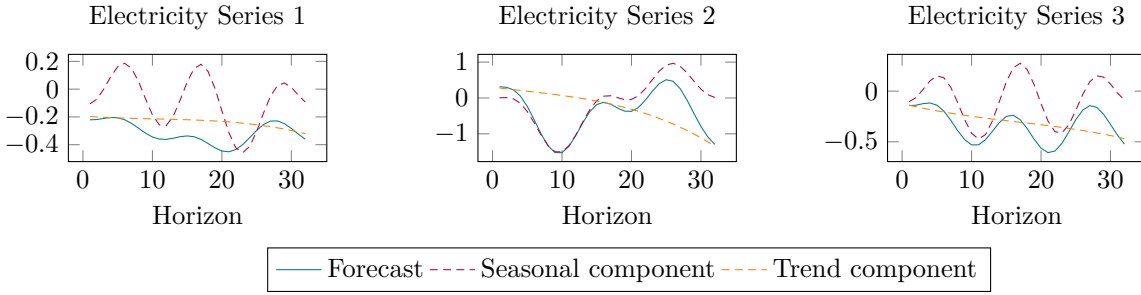

Figure 6: Decomposition of Forecasts into Seasonal and Trend Components for Electricity Dataset

## 6.6 Ablations

To assess the contribution of NFT's key architectural choices, we perform the following ablation studies. Full results are reported in Table 5.

### 6.6.1 Impact of MFT.

In this ablation test, we study the impact of learning from multiple times series vs just a single times series in the NFT framework. Specifically, we evaluate the impact of replacing 2D-DFT with 1D-DFT. We observe that 1D-DFT increases MSE by 1.4x in Exchange, 1.3x in Traffic, 1.2x in Weather, and 1.1x in ETTm1 averaged across all horizons. This highlights the importance of applying DFT across the variable dimension to capture correlations and uncover frequency patterns within and across variables.

### 6.6.2 Impact of Trend Blocks.

We assess NFT variants using only seasonality blocks. Excluding trend blocks increases MSE by 2.2x, 2.1x, 1.5x, and 1.6x on the Exchange, Traffic, Weather, and ETTm1 datasets, respectively, averaged across all horizons. Conversely, in an ablation using only trend blocks, the average MSE increases by 1.4x. These results demonstrate that neither component is sufficient in isolation. This underscores the importance of the doubly residual architecture in jointly capturing both polynomial trend trajectories and periodic seasonal fluctuations.

### 6.6.3 LSTM, SSM, and FC Layers as TCN Alternatives.

We evaluated the performance of NFT when replacing the TCN layers with either Fully Connected (FC) layers, as used in N-BEATS, or LSTM layers, or SSM (State Space Model) layers. Detailed implementations of the LSTM-, SSM-, and FC-based models are provided in Appendix 8.5. TCN layers consistently outperformed the LSTM-, SSM-, and FC-based variants. Replacing TCN with LSTM layers increased the MSE by 1.5x, 1.2x, 1.8x, and 1.9x on the Exchange, Traffic, Weather, and ETTm1 datasets, respectively, averaged across all horizons. Similarly, replacing TCN with FC layers resulted in an increase of 2.3x, 1.6x, 1.4x, and 1.3x in these datasets. Replacing TCN with SSM layers also led to substantial degradation, increasing the MSE by 2.4x, 2.1x, 1.4x, and 1.9x on the Exchange, Traffic, Weather, and ETTm1 datasets, respectively.

### 6.6.4 Spectral vs. Learned Variable Mixing in NFT

To address whether the success of the NFT is due to its spectral properties or simply its role as a global mixer, we compare the 2D-DFT against two alternative architectures:

1. **Linear Mixer:** We replace the 2D-DFT with a 1D-DFT along the time axis followed by a learnable Linear layer over the variable axis shared across all seasonality blocks. This represents an unconstrained global mixer.

2. **Mean:** We apply a 1D-DFT along time and replace the variable-axis transform with a simple mean-pooling operation, which is expanded back to the original dimensions. This represents an approach that ignores inter-variable relationships.

As shown in Table 5, replacing the 2D-DFT with a linear mixer increases MSE by $1.1\times$ to $1.2\times$ across all datasets. The Mean variant results in even greater degradation (up to $1.3\times$ on ETTm1), except on the Exchange dataset. These results demonstrate that the 2D-DFT provides an inductive bias for multivariate time series.

Table 5: Ablation results on four representative datasets, reporting MSE averaged across all forecast horizons. We compare the NFT model to ablated variants: (1) with 1D-DFT to assess the role of inter-series frequency modeling, (2) isolating the contributions of the doubly residual Trend and Seasonality blocks, (3) with LSTM or Fully Connected (FC) or SSM layers replacing the TCN backbone, and (4) comparing 2D-DFT against 1D-DFT and alternative variable mixing strategies (Linear Mixer and Mean).

| Dataset | NFT | NFT 1D-DFT | NFT only Seasonality Block | NFT only Trend Block | NFT based LSTM | NFT based FC | NFT based SSM | Linear Mixer | Varaible Mean |
|---|---|---|---|---|---|---|---|---|---|
| Exchange | 0.329 | 0.461 | 0.716 | 0.467 | 0.493 | 0.756 | 0.807 | 0.350 | 0.310 |
| Traffic | 0.328 | 0.426 | 0.682 | 0.506 | 0.393 | 0.524 | 0.690 | 0.359 | 0.388 |
| Weather | 0.403 | 0.484 | 0.604 | 0.453 | 0.725 | 0.564 | 0.592 | 0.422 | 0.442 |
| ETTm1 | 0.308 | 0.339 | 0.492 | 0.401 | 0.585 | 0.400 | 0.596 | 0.382 | 0.407 |

# 7   Conclusions

In this study, we formulate multivariate time series forecasting as a Multi-Dimensional Fourier Transform (MFT) problem. To the best of our knowledge, this is the first work to apply MFT to model multivariate time series, enabling a more expressive representation of seasonal patterns across both temporal and variable dimensions.

We propose the Neural Fourier Transform (NFT) algorithm to learn the MFT coefficients of unknown future timesteps. By utilizing MFT, NFT comprehensively analyzes time series data across both temporal and variable dimensions, effectively capturing complex temporal dependencies. Additionally, the integration of TCN layers for learning FT coefficients further enhances NFT's forecasting performance.

NFT is evaluated on 7 diverse datasets under standard settings, achieving superior performance compared to 17 SOTA methods. Its effectiveness is particularly evident in datasets with strong seasonality.

In this study, NFT builds on the methodology of N-BEATS. Future work could explore integrating NFT with other transparent forecasting frameworks, like Prophet Zunic et al. (2020), which currently lack multivariate prediction capabilities.

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

# 8 Appendix

## 8.1 Datasets

The experiments were conducted using 7 datasets commonly employed in comparative studies of multiple time series predictions. Each dataset was partitioned and preprocessed following established guidelines Nie et al. (2022); Wu et al. (2022; 2021); Zeng et al. (2023); Zhou et al. (2022b); Raghunath et al. (2020); Batista et al. (2024); Kotecha (2018).

- Hourly **Electricity** consumption of 321 customers from 2012 to 2014[2],

- Daily **Exchange** Rates of eight different countries ranging from 1990 to 2016 Wu et al. (2021)

- Hourly **Traffic** data provided by the California Department of Transportation, detailing road occupancy rates recorded by various sensors located on freeways in the San Francisco Bay area[3]

- The **ETTm1** dataset comprising data gathered from an electric transformer, including load and oil temperature measurements taken every 15 minutes from July 2016 to July 2018 Zhou et al. (2021)

- 12-lead Electrocardiogram (**ECG**) Goldberger et al. (2000)

- 36-lead Electroencephalogram (**EEG**) Obeid & Picone (2016)

- **Weather** dataset includes daily climate summaries from a global station. [4].

---

[2] https://archive.ics.uci.edu/dataset/321/electricityloaddiagrams20112014
[3] https://pems.dot.ca.gov/
[4] https://www.ncei.noaa.gov/products/land-based-station/global-historical-climatology-network-daily

## 8.2 Hyperparameter Tuning

The NFT model comprises two distinct stacks: one dedicated to capturing trends and the other to seasonality. The number of blocks in each stack, Fourier granularity in the seasonality block, and polynomial degree in the trend block were tuned for each dataset to optimize performance. Hyperparameters were selected using a grid search method, balancing model complexity and forecasting accuracy.

Table 6 summarizes the best-performing configurations for each dataset.

Table 6: Best-performing hyperparameter configurations for the NFT model on each dataset. We report the number of blocks per stack, the Fourier granularity used in the seasonality block, and the polynomial degree used in the trend block. All values were selected via grid search based on validation performance.

| Data | Num. Blocks | Fourier Granularity | Poly Degree |
|---|---|---|---|
| Weather | 2 | 2 | 4 |
| Exchange | 2 | 8 | 4 |
| ETTm1 | 2 | 8 | 4 |
| Traffic | 2 | 14 | 4 |
| Electricity | 2 | 16 | 4 |
| EEG | 3 | 8 | 4 |
| ECG | 3 | 8 | 4 |

To ensure fidelity to prior work, we adopted the per-dataset hyperparameter configurations reported as optimal in the original baseline papers. See the full hyperparameter listings in Table 7.

Table 7: Key hyperparameters used for all baseline models. The hyperparameters were copied directly from the original papers. When a value appears in $\{\cdot\}$, it indicates that these dimensions were varied across datasets.

| Model | Hyperparameters |
|---|---|
| DLinear | Epochs = 10 + Early Stopping; |
| TimesNet | Epochs = 10 + Early Stopping; encoder layers $\in \{1, 2\}$; decoder layers = 1; factor = 3; $d_{\text{model}} \in \{20, 32, 64, 256\}$; $d_{\text{ff}} \in \{20, 64, 512\}$; top-$k \in \{1, 3, 5\}$; num_kernels = 6 |
| FiLM | Epochs = 10 + Early Stopping; encoder layers = 2; decoder layers = 1 |
| iTransformer | Epochs = 10 + Early Stopping; $d_{\text{ff}} \in \{20, 32\}$; $d_{\text{model}} \in \{20, 32\}$ |
| TSLANet | Epochs = 10; embedding dim = 128; depth = 2 |
| PatchTST | Epochs = 100 + Early Stopping; encoder layers = 2; decoder layers = 1; $d_{\text{model}} = 16$; $d_{\text{ff}} = 128$; $n_{\text{heads}}$ (default from repo); |
| FRNet | Epochs = 100 + Early Stopping; encoder layers (default); $n_{\text{heads}} = 8$; $d_{\text{model}} = 64$; $d_{\text{ff}} \in \{128, 256\}$; dropout = 0.1; fc_dropout = 0.1; patch_len = 16; stride = 8 |
| ATFNet | Epochs = 10 + Early Stopping; encoder layers = 2; decoder layers = 1 |
| FourierGNN | Epochs = 10 + Early Stopping; feature_size = 140; hidden_size = 8 |
| CycleNet | Epochs = 10 + Early Stopping; cycle = 168 |
| SOFTS | Epochs = 30 + Early Stopping; $d_{\text{model}} = 512$; $d_{\text{core}} = 128$; $d_{\text{ff}} = 512$ |
| SparseTSF | Epochs = 30 + Early Stopping; $d_{\text{model}} = 128$; encoder layers = 2; decoder layers = 1; $d_{\text{ff}} = 2048$ |
| U-Mixer | Epochs = 10 + Early Stopping; encoder layers = 2; decoder layers = 1; $d_{\text{model}} = 64$; $d_{\text{ff}} = 64$ |
| TimeMixer | Epochs = 20 + Early Stopping; $d_{\text{ff}} = 16$; $d_{\text{model}} = 32$ |
| TCN | Epochs = 10; kernel_size = 3; levels = 5 |
| N-BEATS | Epochs = 10; 2 stacks |
| VAR | – |

### 8.2.1 Sensitivity to Fourier Granularity

To assess the robustness of the NFT framework, we conducted a sensitivity analysis on the Fourier granularity parameter $(k)$, which defines the number of harmonics used in the spectral blocks. As detailed in Table 8, the model demonstrates high stability across a wide range of $k$ values.

We observe that the sensitivity of the model is closely tied to the variable dimensionality of the input data. For low-dimensional datasets such as Weather (3 variables) and ETTm1 (7 variables), the MSE remains nearly constant as $k$ increases. This suggests that for datasets with fewer series, the cross-variable frequency patterns are relatively simple and can be captured with low spectral modes.

Conversely, the Electricity dataset (321 variables) shows a more pronounced sensitivity, with performance generally improving as granularity increases from 2 to 16. The high variable count in Electricity introduces intricate inter-series dependencies that require a higher-dimensional spectral basis to model effectively. Crucially, even when $k$ is set higher than the optimal value for simpler datasets, performance does not significantly degrade, indicating that the NFT architecture is robust to over-parameterization in the spectral domain.

Table 8: Sensitivity analysis of the Fourier Granularity parameter ($k$) across three representative datasets. We report the average MSE across all horizons. Results indicate that while high-dimensional datasets like Electricity benefit from higher granularity, low-dimensional datasets like Weather and ETTm1 remain stable even with minimal spectral modes.

| Data | Number of variable | $k = 2$ | $k = 4$ | $k = 6$ | $k = 8$ | $k = 10$ | $k = 12$ | $k = 14$ | $k = 16$ |
|---|---|---|---|---|---|---|---|---|---|
| Weather | 3 | 0.400 | 0.407 | 0.405 | 0.403 | 0.402 | 0.404 | 0.403 | 0.403 |
| ETTm1 | 7 | 0.332 | 0.314 | 0.310 | 0.308 | 0.311 | 0.309 | 0.308 | 0.308 |
| Electricity | 321 | 0.127 | 0.110 | 0.103 | 0.095 | 0.107 | 0.104 | 0.114 | 0.091 |

## 8.3 MAE Results

Table 9: MAE results across models and datasets. Best results are in bold and second-best are underlined.

| Data | Horizon | NFT | TimeMixer | U-Mixer | SparseTSF | SOFTS | CycleNet | FourierGNN | ATFNet | FRNet | TSLANet | iTransformer | FiLM | TimesNet | PatchTST | DLinear | TCN | N-BEATS | VAR |
|---|---|---|---|---|---|---|---|---|---|---|---|---|---|---|---|---|---|---|---|
| Exchange | 96 | **0.173** | 0.227 | 0.206 | 0.201 | 0.257 | 0.228 | 0.475 | 0.230 | 0.200 | 0.201 | 0.206 | 0.240 | 0.234 | 0.205 | 0.203 | 0.430 | 0.720 | 0.946 |
| | 192 | **0.290** | 0.331 | 0.321 | 0.295 | 0.359 | 0.331 | 0.756 | 0.370 | 0.300 | 0.299 | 0.299 | 0.320 | 0.344 | 0.299 | 0.293 | 0.740 | 0.860 | 0.947 |
| | 336 | 0.405 | 0.646 | 0.421 | 0.406 | 0.507 | 0.469 | 0.826 | 0.450 | 0.420 | 0.417 | 0.417 | 0.410 | 0.448 | **0.397** | 0.414 | 0.770 | 0.820 | 1.461 |
| | 720 | **0.601** | 0.770 | 0.603 | 0.678 | 0.788 | 0.800 | 0.911 | 0.620 | 0.710 | 0.739 | 0.691 | 0.710 | 0.746 | 0.714 | 0.607 | 1.720 | 1.740 | 1.472 |
| Traffic | 1 | **0.190** | 0.193 | 0.223 | 0.381 | 0.197 | 0.194 | 0.401 | 0.219 | 0.250 | 0.182 | 0.260 | 0.790 | 0.410 | 0.230 | 0.250 | 0.220 | 0.200 | 1.646 |
| | 16 | 0.270 | 0.259 | 0.294 | 0.408 | **0.220** | 0.285 | 0.455 | 0.396 | 0.320 | 0.276 | 0.410 | 0.390 | 0.330 | 0.380 | 0.400 | 0.330 | 0.280 | 0.582 |
| | 32 | 0.330 | 0.372 | **0.306** | 0.425 | 0.331 | 0.275 | 0.487 | 0.412 | 0.350 | 0.290 | 0.460 | 0.400 | 0.400 | 0.400 | 0.420 | 0.331 | 0.340 | 1.738 |
| | 48 | 0.410 | 0.290 | 0.313 | 0.436 | 0.242 | 0.285 | 0.524 | 0.421 | 0.360 | 0.300 | 0.480 | 0.505 | 0.350 | 0.410 | 0.440 | 0.390 | 0.500 | 1.182 |
| ETTm1 | 96 | **0.310** | 0.376 | 0.340 | 0.349 | 0.368 | 0.600 | 0.608 | 0.375 | 0.340 | 0.343 | 0.475 | 0.410 | 0.480 | 1.110 | 0.378 | 0.387 | 0.396 | 0.387 |
| | 192 | **0.307** | 0.411 | 0.360 | 0.370 | 0.391 | 0.310 | 0.621 | 0.387 | 0.365 | 0.365 | 0.496 | 0.630 | 0.530 | 1.112 | 0.340 | 0.391 | 0.436 | 0.409 |
| | 336 | 0.378 | 0.442 | 0.382 | 0.389 | 0.420 | **0.330** | 0.854 | 0.411 | 0.389 | 0.386 | 0.537 | 0.740 | 0.630 | 0.968 | 0.365 | 0.434 | 0.484 | 0.437 |
| | 720 | 0.439 | 0.481 | 0.411 | 0.425 | 0.459 | **0.370** | 0.889 | 0.450 | 0.423 | 0.421 | 0.561 | 0.720 | 0.960 | 0.886 | 0.381 | 0.465 | 0.530 | 0.479 |
| Weather | 96 | **0.491** | 0.581 | 0.504 | 0.677 | 0.561 | 0.652 | 0.869 | 0.561 | 0.502 | 0.659 | 0.802 | 0.878 | 0.657 | 0.751 | 0.493 | 0.501 | 0.497 | 0.790 |
| | 192 | **0.510** | 0.732 | 0.534 | 0.818 | 0.655 | 0.781 | 0.948 | 0.716 | 0.698 | 0.933 | 0.946 | 0.961 | 0.782 | 0.900 | 0.531 | 0.540 | 0.521 | 0.954 |
| | 336 | **0.498** | 0.626 | 0.518 | 0.708 | 0.596 | 0.679 | 0.916 | 0.648 | 0.612 | 0.757 | 0.830 | 0.784 | 0.664 | 0.773 | 0.516 | 0.524 | 0.516 | 0.882 |
| | 720 | 0.536 | 0.654 | **0.529** | 0.699 | 0.599 | 0.686 | 0.899 | 0.664 | 0.659 | 0.767 | 0.826 | 0.786 | 0.701 | 0.762 | 0.521 | 0.537 | 0.562 | 0.848 |
| ECG | 10 | **0.305** | 0.333 | 0.324 | 0.371 | 0.323 | 0.339 | 0.608 | 0.370 | 0.390 | 0.400 | 0.525 | 0.708 | 0.510 | 0.560 | 0.610 | 0.510 | 0.540 | 1.001 |
| | 25 | **0.375** | 0.514 | 0.406 | 0.527 | 0.411 | 0.445 | 0.783 | 0.443 | 0.520 | 0.460 | 0.640 | 0.788 | 0.580 | 0.650 | 0.640 | 0.600 | 0.590 | 1.183 |
| | 50 | 0.564 | 0.489 | **0.417** | 0.640 | 0.491 | 0.572 | 0.821 | 0.514 | 0.624 | 0.486 | 0.753 | 0.888 | 0.654 | 0.729 | 0.612 | 0.709 | 0.611 | 0.698 |
| | 100 | **0.481** | 0.746 | 0.507 | 0.696 | 0.556 | 0.584 | 0.857 | 0.613 | 0.933 | 0.490 | 0.764 | 0.848 | 0.800 | 0.749 | 0.700 | 0.730 | 0.730 | 0.773 |
| EEG | 1 | **0.101** | 0.234 | 0.157 | 0.191 | 0.133 | 0.154 | 0.298 | 0.186 | 0.166 | 0.240 | 0.155 | 0.321 | 0.240 | 0.250 | 0.190 | 0.165 | 0.250 | 0.868 |
| | 10 | **0.194** | 0.312 | 0.287 | 0.358 | 0.215 | 0.329 | 0.402 | 0.345 | 0.380 | 0.360 | 0.375 | 0.420 | 0.390 | 0.410 | 0.380 | 0.370 | 0.480 | 0.792 |
| | 25 | **0.260** | 0.378 | 0.335 | 0.416 | 0.263 | 0.391 | 0.581 | 0.421 | 0.420 | 0.405 | 0.420 | 0.498 | 0.430 | 0.460 | 0.430 | 0.435 | 0.540 | 0.894 |
| | 25 | **0.287** | 0.465 | 0.396 | 0.475 | 0.311 | 0.501 | 0.648 | 0.437 | 0.493 | 0.513 | 0.482 | 0.536 | 0.491 | 0.584 | 0.459 | 0.542 | 0.601 | 0.936 |
| Electricity | 1 | 0.146 | 0.533 | 0.184 | 0.158 | **0.129** | 0.137 | 0.678 | 0.170 | 0.180 | 0.150 | 0.170 | 0.750 | 0.690 | 0.500 | 0.170 | 0.410 | 0.220 | 1.353 |
| | 16 | 0.216 | 0.206 | 0.220 | 0.240 | **0.190** | 0.199 | 0.771 | 0.230 | 0.221 | 0.223 | 0.270 | 0.260 | 0.690 | 0.690 | 0.250 | 0.770 | 0.630 | 1.110 |
| | 32 | 0.136 | **0.128** | 0.167 | 0.259 | 0.206 | 0.214 | 0.914 | 0.250 | 0.240 | 0.240 | 0.300 | 0.270 | 0.710 | 0.700 | 0.270 | 0.900 | 0.680 | 0.597 |

Table 9 presents the results measured using the mean absolute error (MAE) metric Willmott & Matsuura (2005). The NFT model achieves average improvement percentages of **30.26%** in EEG, **3.75%** in Exchange, and **2.15%** in Weather.
The improvements were calculated by comparing the NFT model to the best-performing baseline for each horizon, with the MAE improvement percentages averaged across all time horizons for each dataset.

## 8.4 Synthetic Data Generation

To evaluate the relationship between seasonality and NFT performance, we generated a comprehensive set of synthetic time series data with varying combinations of trend and seasonality strengths. Each time series was created by combining three components: a linear trend, a sinusoidal seasonality pattern with randomized phases and adjustable amplitude, and additive Gaussian noise. The seasonality amplitude ranged across predefined values, while the trend amplitude was scaled relative to the time index. For each series, we calculated the proportions of variance attributed to trend, seasonality, and noise to quantify their relative contributions.

The generation process ensured diversity in seasonal dominance, measured using the Seasonal Proportion (SP). This allowed us to group time series with similar SP values for robust model performance comparisons. Each dataset consisted of multivariate time series with five variables and 1,000 time steps, ensuring sufficient length for meaningful forecasting analysis.

### 8.5 LSTM, SSM, and Fully Connected Layers as Alternatives to TCN

In this ablation study, we examined the effects of substituting the Temporal Convolutional Network (TCN) layers in our original model with Long Short-Term Memory (LSTM) layers, State Space Model (SSM) layers, and Fully Connected (FC) layers, similar to the structure used in the N-BEATS model.

The modified LSTM-based architecture consisted of two layers, each with 50 hidden units, while maintaining the same input configuration as the original TCN-based setup. The FC variant followed the N-BEATS design, with four layers of 256 units each. For the SSM variant, we implemented a lightweight Mamba-inspired encoder that combines depthwise convolutional mixing with selective state-space cells. This design allows linear-time recurrence while capturing long-range dependencies.

All three variants were trained under the same settings as the original NFT model for fair comparison. We evaluated their performance against the TCN-based NFT using the MSE metric. Across datasets, the TCN consistently outperformed the LSTM-, SSM-, and FC-based models, as detailed in Section 6.6.

### 8.6 Comparison with Foundation Model

To further validate the robustness and specialized forecasting capabilities of NFT, we compare its performance against Chronos Ansari et al. (2024), a Time Series Foundation Model (TSFM). We evaluate the `chronos-t5-base` variant in a zero-shot setting. In this configuration, the model generates forecasts for our target datasets without any dataset-specific fine-tuning, leveraging only the general temporal patterns learned during its massive pre-training on billions of data points.

As demonstrated in Table 10, NFT outperforms the zero-shot Chronos model across all benchmarks. We identify two primary reasons for this performance gap:

1. **Multivariate Modeling:** Chronos is natively a univariate model that processes each variable in isolation. In contrast, NFT utilizes a 2D-DFT to explicitly capture inter-variable correlations. This variate-mixing is essential for high-dimensional datasets such as *Electricity*, where the signals are highly interdependent.

2. **Horizon Constraints:** Chronos uses an autoregressive transformer architecture that is primarily optimized for shorter forecasting horizons. Performance tends to degrade for prediction lengths significantly exceeding 64 steps, as the model is susceptible to error accumulation during the recursive decoding process. Conversely, NFT operates in the frequency domain to capture global periodicities, allowing it to maintain stability across the extended horizons (up to 720 steps) required by the evaluated benchmarks.

### 8.7 Extended Spectral Sensitivity Analysis

Figures 7 and 8 provide additional 2D-DFT magnitude spectra for the Electricity and ECG datasets under nine different random variable permutations. These visualizations support the discussion in Section 6.3.

Table 10: MSE comparison between the proposed NFT model and the Chronos-Base foundation model. Results for Chronos are reported in a zero-shot setting.

| Dataset | Horizon | NFT | Chronos (Zero-shot) |
|---|---|---|---|
| Electricity | 1 | **0.051** | 0.077 |
| | 16 | **0.080** | 0.215 |
| | 32 | **0.141** | 0.266 |
| Exchange | 96 | **0.060** | 0.380 |
| | 192 | **0.164** | 0.862 |
| | 336 | **0.422** | 1.284 |
| | 720 | **0.670** | 2.677 |
| Traffic | 1 | **0.200** | 0.373 |
| | 16 | **0.320** | 0.997 |
| | 32 | **0.390** | 1.163 |
| | 48 | **0.400** | 1.316 |
| ETTm1 | 48 | **0.270** | 1.161 |
| | 96 | **0.241** | 1.359 |
| | 192 | **0.320** | 1.533 |
| | 336 | **0.400** | 1.578 |
| Weather | 96 | **0.364** | 1.133 |
| | 192 | **0.383** | 1.684 |
| | 336 | **0.384** | 1.510 |
| | 720 | **0.481** | 1.475 |
| ECG | 10 | **0.172** | 0.477 |
| | 25 | **0.258** | 0.968 |
| | 50 | **0.400** | 1.513 |
| | 100 | **0.393** | 1.862 |
| EEG | 1 | **0.075** | 0.119 |
| | 10 | **0.159** | 0.491 |
| | 25 | **0.238** | 0.687 |
| | 50 | **0.297** | 0.964 |

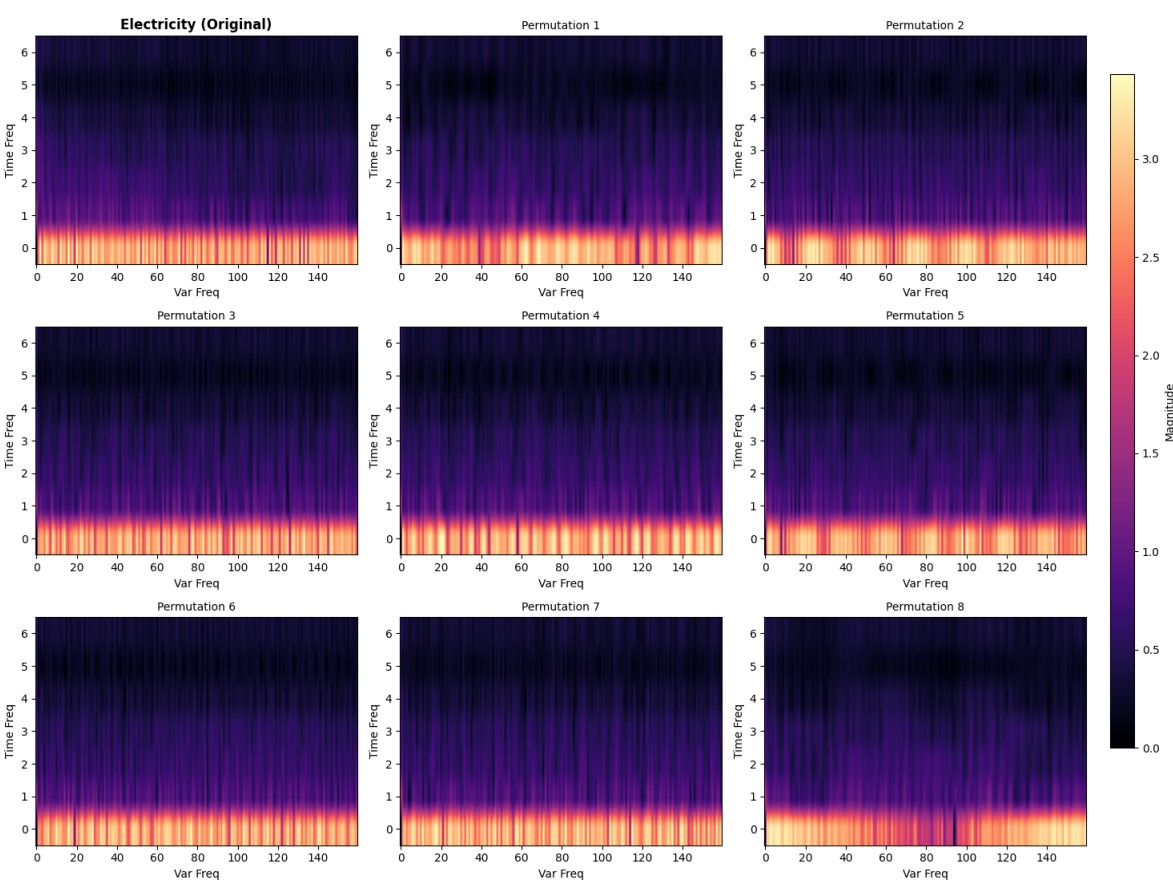

Figure 7: 2D-DFT Magnitude Spectra for Electricity under 9 random permutations.

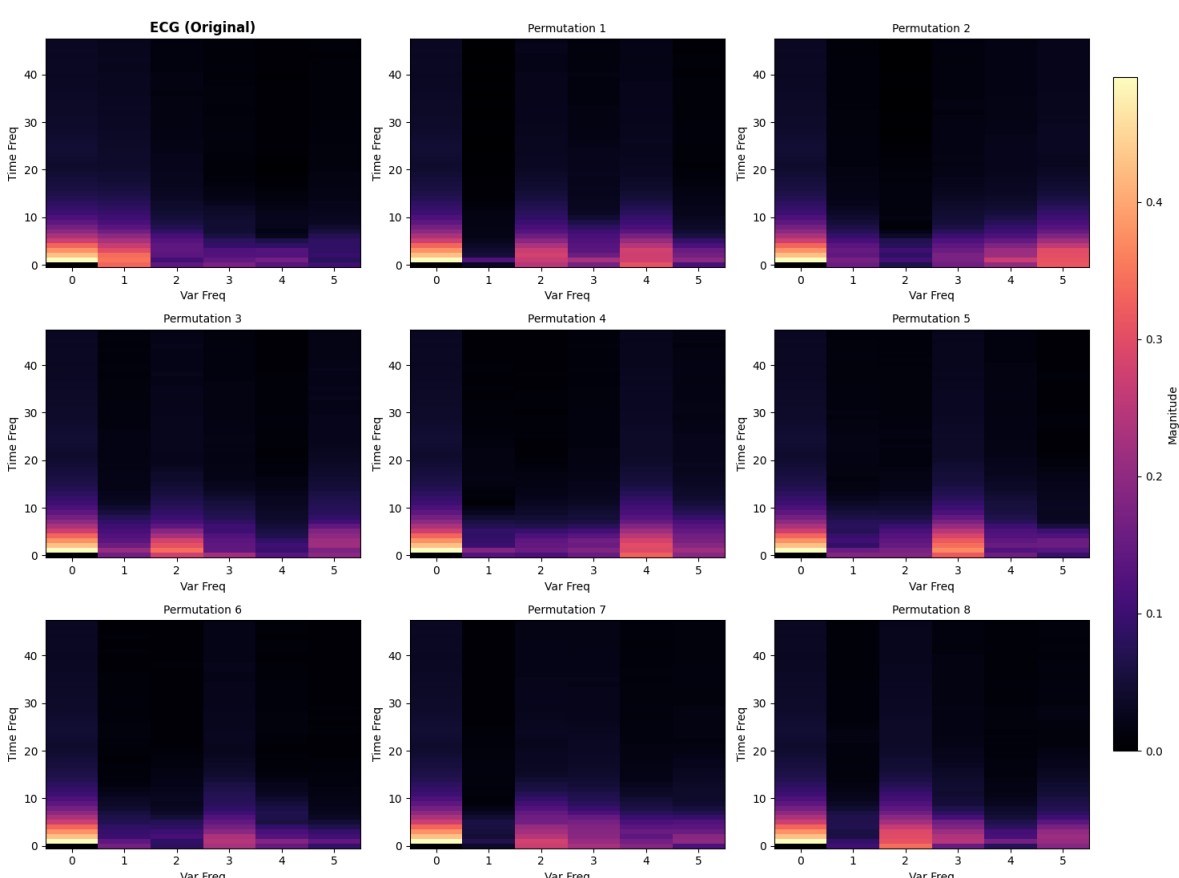

Figure 8: 2D-DFT Magnitude Spectra for ECG under 9 random permutations.

