# OpenReview forum: "Neural Fourier Transform for Multiple Time Series Prediction"
_TMLR — Accepted by TMLR_

### Review · Reviewer_abZ4 · 2025-12-01

**Summary Of Contributions:**

The paper proposes a deep neural architecture for forecasting multivariate time series with seasonal components. From a fixed-length input window, the model learns 2D-DFT coefficients and polynomial trend coefficients, which are used within an iterative residual-refinement framework. The forecast is reconstructed from these learned coefficients via an inverse 2D-DFT and a polynomial trend decoder. The model demonstrates superior performance on several benchmark datasets compared to state-of-the-art baselines, in the settings where all the models were trained for 10 epochs. Some important details about the experimental setup are missing. Additional experiments on synthetic data with varying levels of seasonality further illustrate the conditions under which the model is most effective.

**Audience:**

Yes

**Audience Explanation:**

NFT can work well because it directly models seasonality using a Fourier basis, which efficiently captures the dominant periodic structure present in many multivariate time-series datasets. Combined with the TCN encoder, it seems it has the potential to learn useful coefficients quickly and reconstruct accurate forecasts with relatively little training.

**Broader Impact Concerns:**

no ethics concerns

**Claims And Evidence:**

No

**Claims Explanation:**

Main concern:
1. The empirical comparison is performed under a fixed 10-epoch training budget for all models, rather than training each baseline to convergence or using their recommended training schedules. This setup favors architectures that converge quickly (such as TCN-based and Fourier-based models) and disadvantages architectures that typically require longer training (e.g., Transformers, N-BEATS, or SSMs). As a result, the reported performance does not necessarily reflect the best achievable MSE of the baseline models.

2. Appendix 8.2 contains only  NFT configs. There is no information on whether baselines were finetuned.

3. There are no details on how the data was split into train/test/val. Will the results be stable wrt different test/train data resampling?

Minor concerns

4. 10 seeds are typically insufficient for reliable 95% confidence estimation (Table 3)

5. C_back, C_fore could be more explicitly mentioned in the text. It seems C_back and C_fore are linked - aren’t they supposed to be just shifted in time relatively to each other? Is there a reason why F_H starts from 0 time and not from T+1?

6. Such an inferior performance in Figure 2 of NFT is surprising, given that there is an explicit linear component in the polynomial part, and the data has just a linear trend with noise. How can this be explained?

**Requested Changes:**

Please address the concerns 1-6 raised before.
For 1, it would make sense to let the other models converge to compare them in terms of MSE, FLOPs, parameter count, and training time.

Additional minor comments:
- Figure 2 has intersecting values in the x-axis ticks.
- Figure 4, is it correct that the dominant time frequency corresponds to the constant signal? The image would benefit from showing the log of the Magnitude.

---

> ### Author Response · Authors · 2025-12-11
>
> 1. There seems to be a typo in the paper. We ran the authors official code with the reported number of optimal epochs and hyperparameters, ensuring consistency with the original methodology. Most baselines optimal number of epochs as reported in their paper was 10 epochs. To clarify this, we have added a table in Section 8.2 (Table 7 of the Appendix) that includes the full optimal configurations for each baseline model.
>
> 2. All baseline models were trained using their official implementations and the reported optimal hyperparameters, without any additional fine-tuning (our model is also not fine tuned). Our objective was to maintain strict fairness and reproducibility by adhering exclusively to the configurations provided by the authors. We have added clarifying information in Section 5.3 and Appendix 8.2.
>
> 3. The data split follows the standard protocol commonly used in prior forecasting works, with a 7:1:2 train/validation/test division (TimesNet, etc.). This is clarified in Section 5.3.
> Since this is temporal data, we employ non-shuffled splits to prevent data leakage, ensuring no overlap between segments and no shuffling across time. Specifically, if the data consists of T time steps, the first 0.7T are for training, the next 0.1T for validation, and the final 0.2T for testing (as done in TimesNet, etc.).
> We added experiments with partitionings of (6:1:3 and 8:1:1) to Appendix 8.6.
> Given that this is time-series data, naive k-fold cross-validation is not appropriate, which is why we adopt this temporal partitioning approach instead.
>
> 4. To improve the precision of the estimated mean, we have increased the number of runs to N = 20 for. We now report 95% CIs over 20 runs. The findings remain consistent with the original results.
>
> 5.1. “C_back’’ and “C_fore’’ aren’t time-shifted versions of one another. They are 2 separate sets of Fourier coefficients that the model learns in parallel:
>
> C_back represents coefficients that reconstruct the observed past X.
>
> C_fore represents the coefficients that describe the unknown future y.
>
> Given the past sequence X, each block outputs both C_back and C_fore. Using these, the model reconstructs the past (via C_back) and predicts the future (via C_fore).
>
> NFT is a stacked residual architecture. After each block reconstructs the past, the next block receives the residual: X - reconstructed past. This residual contains the part of the signal that hasn’t been captured by the learned structure. The future predictions are accumulated across blocks.
>
> The reason we model the past and the future in the same spectral space is that the residual from the past reconstruction serves as the informative component for predicting the future. For example, in a seasonal time series, the model learns the seasonal component for both past and future. Subtracting the reconstructed seasonal component from the past leaves the trend or irregular component, which subsequent blocks learn.
>
> Thus, C_back and C_fore play different roles and aren’t temporal shifts of one another.
> We have clarified this in section 4.5.
>
> 5.2. Although the forecast window in absolute time spans indices [T+1,T+H], the Fourier basis operates in relative coordinates, so the window is mapped to [0,H−1]. This matches the mathematical definition of Fourier modes and ensures that the learned spectral representation depends only on the structure inside the forecast window, not its absolute location in the sequence.
>
> 6. The linear component is not the primary source of NFT’s expressivity; the model’s core contribution lies in its seasonality decomposition. We utilize a basic linear component without extensive tuning, which explains the performance on purely linear data. In cases where data exhibits a linear trend with noise, the model’s advantages are less pronounced, as the primary benefit of the seasonal decomposition is not fully utilized.
>
> Fig2: Fixed
>
> Fig4:  We updated the paper to show the log of the Magnitude.
> The dominant components in the time–frequency spectrum correspond to the low-frequency terms (e.g., 0–10 in ECG and 0–1 in Electricity).
> The purpose of Fig. 4 isn’t to highlight temporal oscillations but to illustrate why applying a 2DDFT is important. The figure shows that the spectral structure varies strongly across the variable dimension: different variables exhibit different dominant frequencies and different spectral magnitudes. This variation is precisely what motivates applying the Fourier transform across the variable axis, enabling NFT to capture cross-variable interactions in a shared spectral space.

---

> > ### Comment · Reviewer_abZ4 · 2026-03-10
> >
> > Dear Authors,
> > thank you for addressing my comments. For some reason, I could not see your reply posted on Dec 11, it became visible only last week.
> >
> > A couple of remaining questions related to the model comparison:
> > 1. In Table 2 of the manuscript, Time Mixer x Weather experiment reports MSE numbers {0.573, 0.990, 0.696, 0.747}  for horisons {96, 192, 336, 720}  whereas in the TimeMixer paper [https://arxiv.org/pdf/2405.14616], the results in the Table 13 are {0.163, 0.208, 0.251, 0.339}. For Time Mixer x ETTm1 the results reported for horizons {96, 192, 336} are exactly the same as in TimeMixer paper.  Are the splits used are random, or are the following previous works? In original TimeMixer paper, the hyperpapemeters reported for Weather are a bit different (e.g. batch size is 128), and d_model varies dependent on the dataset  - could it be the reason?  While one can expect variablility due to the data split,  iTransformer results are  significantly worse (up to 7 times) than reported in published papers, e.g. (https://arxiv.org/pdf/2404.14197) - for Weather dataset. Could you comment on that?
> >
> > 2. From the revision “All datasets were partitioned using the standard 7:2:1 train/validation/test split commonly adopted
> > in recent forecasting benchmarks Wu et al. (2022); Zeng et al. (2023).”  - Is 7:2:1 a misprint? Later in the text  it is  “7:1:2” for the train/val/test for most of the datasets. Also splits for ETTm1 in the cited literature the split was 6:2:2 - is it the case?

---

> > > ### Author Response · Authors · 2026-03-10
> > >
> > > Thank you for raising these points.
> > >
> > > 1. The discrepancy is not caused by random splitting, as our experiments use fixed temporal train/validation/test splits without shuffling. Rather, we believe the main reason is that, unlike ETT, the “Weather” benchmark is not fully uniform across papers. In our reproduction effort, we found several different Weather sources used in the literature. In our manuscript, the Weather dataset is taken from the NOAA website (Appendix A and Footnote 4), specifically station AEM00041217. As a result, direct comparison with papers that use a different Weather source or station may be imperfect. This also explains why ETTm1 aligns much more closely with previously reported results: that benchmark is considerably more standardized across works.
> > >
> > > 2. The “7:2:1” wording in the revision is a typo. The actual split used in our main experiments is 7:1:2 (70% train, 10% validation, 20% test), and we corrected this throughout the manuscript. More generally, in our experiments we used a uniform chronological 7:1:2 train/validation/test split across datasets, including ETT. While some previous papers use different protocols for ETT, we intentionally chose a single split policy to keep the benchmark setup consistent across datasets and to make our results easier to reproduce.

---

### Review · Reviewer_fhoQ · 2026-01-15

**Summary Of Contributions:**

The authors propose a new method for multivariate time series forecasting which formulates it as a multidimensional Fourier transform problem. For this, they propose the neural Fourier transform, which leverages a deep learning model to predict MFT coefficients for future time series values. They show the effectiveness of their method across datasets and compare it to an extensive set of baselines.

**Additional Comments:**

N/A

**Audience:**

Yes

**Audience Explanation:**

I believe that the contributions offered by this work are significant enough to merit interest from the TMLR audience.

**Claims And Evidence:**

Yes

**Claims Explanation:**

There are extensive experiments presented, both across a large number of datasets with widely varying domains, as well as comparison to a large number of alternative SOTA time-series modeling methods.

**Requested Changes:**

N/A

---

### Review · Reviewer_tFZ4 · 2026-02-07

**Summary Of Contributions:**

Summary Of Contributions

The paper proposes the Neural Fourier Transform (NFT), a novel deep learning architecture for multivariate time series forecasting. The core contribution is the formulation of the forecasting task as a Multi-dimensional Fourier Transform (MFT) problem. Unlike existing spectral methods that apply 1D-DFT solely along the temporal axis, NFT applies 2D-DFT across both the time and variate dimensions to capture global inter-variable dependencies and synchronization.The model adopts a residual stacking architecture composed of seasonality and trend blocks. The seasonality block uses a Temporal Convolutional Network (TCN) backbone to predict Fourier coefficients in the frequency domain, which are then mapped back to the time domain via inverse 2D-DFT. The trend block models low-frequency movements using polynomial fitting. The authors evaluate NFT on 7 standard benchmarks against 17 baselines, including recent Transformers like iTransformer and PatchTST, and achieve state-of-the-art results, particularly on datasets with strong periodicity.

Key Strengths:

The introduction of 2D-DFT offers a new way to model cross-variable correlations as spectral structures, distinguishing it from spatial-temporal GNNs or Attention-based methods.

The method demonstrates significant performance gains on strongly periodic datasets (e.g., Electricity, Traffic, ECG), outperforming current SOTA models.

Ablation studies convincingly demonstrate the superiority of the TCN backbone over Transformers or LSTMs for coefficient learning within this framework.

Key Weaknesses:

The use of FFT implies order sensitivity, yet ablation studies show the model is robust to variable permutation. This suggests the "spectral" interpretation of the variate dimension may be theoretically flawed, as the model treats the variate axis as a set rather than a sequence.

The performance gain is marginal on datasets lacking strong seasonality (e.g., Exchange), indicating the method is highly specialized rather than universally superior.

The model demonstrates significant sensitivity to hyperparameter configurations across different datasets. For instance, the optimal "Fourier Granularity" (number of modes) varies drastically, requiring a setting of 14 for the Traffic dataset but only 2 for the Weather dataset.

**Audience:**

Yes

**Audience Explanation:**

The findings of this paper would be of interest to a broad section of the TMLR audience, particularly those focused on time series analysis, signal processing, and frequency-domain learning.The interest stems from several aspects:

The paper revitalizes the use of Multi-dimensional Fourier Transforms (MFT) in a deep learning context. By demonstrating that a 2D-DFT approach can outperform complex Transformer architectures, it provides a valuable reference for researchers looking for more computationally efficient and mathematically grounded alternatives to attention mechanisms.

The community focused on applied forecasting will find the extensive empirical results (17 baselines across 7 benchmarks) highly useful for understanding the current performance ceiling in the field. The achievement of new SOTA results on classic datasets like Electricity and Traffic makes this a significant contribution to the forecasting "leaderboard."

The methodology of treating the variable dimension as a spectral axis offers a provocative new perspective on multivariate dependency. Even with the noted theoretical discussion regarding variable ordering, the empirical success of this approach will likely inspire further research into permutation-invariant frequency transforms or graph-frequency analysis.

The success of the additive decomposition (Seasonality + Trend) within a high-performance neural network is of interest to practitioners in "high-stakes" domains like healthcare (ECG) and energy management, where understanding the components of a prediction is as crucial as the accuracy itself.

**Broader Impact Concerns:**

There are no significant ethical concerns that would require a mandatory Broader Impact Statement for this submission.

**Claims And Evidence:**

Yes

**Claims Explanation:**

The claims made in this submission are supported by extensive and generally convincing empirical evidence. The authors provide an evaluation of the proposed Neural Fourier Transform (NFT) model across several dimensions:

The model is tested on 7 diverse real-world datasets and compared against a wide array of 17 baselines, including the most recent state-of-the-art Transformer-based models (e.g., iTransformer, PatchTST). The performance gains, especially in strongly periodic datasets like Electricity and ECG, are statistically significant and clearly presented.

The ablation studies are well-designed to justify the core components of the architecture. Specifically, the comparison between 2D-DFT and 1D-DFT provides direct evidence for the benefit of cross-variable spectral modeling. The comparison of different backbones (TCN, LSTM, Transformer) also supports the choice of TCN as an efficient coefficient learner.

The decomposition analysis (visualizing seasonality and trend components) and the stability tests across different temporal splits and lookback windows further strengthen the credibility of the claims.

Note:

While the evidence for "performance" is clear, there is a slight tension between the theoretical motivation and the evidence regarding the variate dimension. The authors claim that NFT captures spectral structures across variables, yet the "Permutation" experiment shows the model is largely indifferent to variable ordering. While this does not invalidate the predictive accuracy, it invites further scrutiny regarding the underlying mechanism. However, within the scope of TMLR's criteria for supported claims, the empirical results sufficiently back the primary assertion that NFT is a superior forecasting tool.

**Requested Changes:**

1. Critical for Acceptance: Addressing the Variate Dimension Paradox
The core innovation—applying DFT to the variate dimension—is theoretically sensitive to the ordering of variables. However, Table 4 shows that random permutation of variables has negligible impact on performance. This creates a significant gap between the theoretical motivation and the experimental findings.

Required Action: The authors must provide a rigorous discussion or additional experiments to clarify why an order-dependent operator (FFT) is effective on an unordered set (variables).

Specific Experiment: Compare the Variate-DFT with a standard learnable Linear layer (acting as a global mixer) and a Permutation-Invariant mixer (such as a simple Mean-Pooling or Set-Transformer block). If a Linear layer performs similarly to the Variate-DFT, the "Spectral" interpretation of the variate dimension must be tempered, and the motivation should be reframed toward "parameter-efficient global mixing."

2. Critical for Acceptance: Detailed Hyperparameter Analysis
The paper mentions that the "Fourier Granularity" (top-k modes) varies significantly across datasets (e.g., 14 for Traffic vs. 2 for Weather).

Required Action: Provide a sensitivity analysis (e.g., a line plot) showing how performance (MSE) changes with different values of Fourier Granularity for at least two representative datasets. This will help readers understand the robustness of the model and the effort required for tuning.

3. Strengthening the Work: Comparison with Foundation Models
While the current 17 baselines are comprehensive for supervised learning, the landscape of time series forecasting has shifted towards Foundation Models.

Proposed Adjustment: A comparison with at least one recent Time Series Foundation Model (e.g., Chronos, TimeGPT, or Moirai) in a Zero-shot or Fine-tuned setting would greatly enhance the paper's impact. It would demonstrate whether NFT’s specialized architecture remains superior to large-scale pre-trained models in domain-specific tasks.

4. Strengthening the Work: Ablation of the Trend Block
While the seasonality block is thoroughly analyzed, the contribution of the Trend Block (polynomial fitting) is less clear.

Proposed Adjustment: Include a brief ablation result showing the performance of NFT with and without the Trend Block, especially on datasets like Exchange where the trend might be more dominant than seasonality.

5. Strengthening the Work: Visualizing Variate-Spectral Features
Figure 4 is excellent. To further strengthen the "Synchronization" argument, it would be beneficial to show the 2D-spectrum of a dataset after permutation. If the spectrum changes drastically but the MSE does not, it reinforces the need for the theoretical clarification requested in point 1.

---

> ### Author Response · Authors · 2026-02-16
>
> We thank the reviewer for their insightful and constructive feedback.
>
> 1, 5. Variate-dimension paradox. We expanded Sec. 6.4 (“Robustness to Variable Ordering”) to clarify the protocol and provide a rigorous discussion. In our permutation study, we permute the variable order before both training and inference (10 independent permutations), so the model is trained end-to-end under a consistent (but random) indexing. Table 4 reports mean MSE ± SD across permutations and confirms high stability with very low variance (e.g., Traffic 0.328±0.003; Electricity 0.091±0.008). We added Fig. 5, illustrating that permutation can visibly alter the 2D spectrum, yet dominant structures persist (e.g., the low variate-frequency peak in ECG), which aligns with the unchanged MSE. Conceptually, unlike CNNs that depend on local adjacency, the DFT is a global operator: each variate-frequency coefficient is a weighted sum over all variables. Therefore, permuting variables primarily reindexes/redistributes spectral energy across the variate-frequency grid rather than destroying information. During training, NFT learns to map the resulting (permuted) spectral representation to the correct forecast, explaining the observed permutation robustness.
>
> 1.2 Linear global mixer + permutation-invariant mixer. As requested, Sec. 6.6.4 and Table 5 include (i) Linear Mixer (1D-DFT over time + learned linear mixing over variables shared across blocks) and (ii) Variable Mean pooling (permutation-invariant). Linear Mixer recovers part of the gain but remains worse than 2D-DFT on most datasets (e.g., Traffic 0.359 vs 0.328; Weather 0.422 vs 0.403; ETTm1 0.382 vs 0.308). Mean pooling generally degrades further (Traffic 0.388; Weather 0.442; ETTm1 0.407), with Exchange being the main exception where mean pooling is competitive, suggesting weaker cross-variable structure in that dataset.
>
> 2. Hyperparameter sensitivity (Fourier granularity, k). We added App. 8.2.1 (Table 7), reporting MSE as a function of k on representative datasets (Weather, ETTm1, Electricity). The results exhibit a broad stability region: Weather and ETTm1 are nearly flat across a wide range of k, while Electricity improves with larger k. This suggests a clear link between dimensionality and the needed spectral granularity—high-dimensional datasets such as Electricity (321 variables) benefit from higher k. In contrast, low-dimensional datasets like Weather (3) and ETTm1 (7) remain robust even with very few spectral modes.
>
> 3. Foundation model comparison. We added Sec. 8.7 and Table 11 comparing NFT to Chronos (chronos-t5-base) in a zero-shot setting. NFT outperforms Chronos across benchmarks/horizons. We discuss likely reasons: 1. Chronos is univariate-by-design (variables processed independently) and 2. Can degrade on long horizons due to autoregressive forecasting, whereas NFT learns continuous multivariate mappings and explicitly models inter-variable structure via 2D-DFT. Due to space constraints, we initially placed this comparison in the appendix; however, we are happy to move it to the main results table by replacing one of the existing baselines.
>
> 4. Trend block ablation. Table 5 now reports “only Seasonality” and “only Trend” variants. Removing the trend stack degrades performance, and trend-only is also inferior, supporting that the trend and seasonality stacks are complementary.

---

> > ### Comment · Reviewer_tFZ4 · 2026-03-03
> > **Acknowledgement of Revision and Request for Further Clarification**
> >
> > I acknowledge the authors' efforts in revising the manuscript and providing responses to my initial concerns.
> >
> > Specifically, the authors have provided a theoretical explanation for the Variate-dimension paradox and included a comparative experiment with a Linear global mixer (Table 5). While these results provide some clarification on the 2D-DFT architecture, my final assessment of the paper’s contribution remains contingent upon the resolution of the broader experimental concerns.
> >
> > I am particularly closely monitoring the discussion regarding the 10-epoch training budget raised by Reviewer abZ4. Since the paper’s primary claim is based on its state-of-the-art (SOTA) performance, it is essential to ensure that the empirical comparison is fair and that the baselines have been trained to sufficient convergence.
> >
> > I look forward to the authors' detailed response to Reviewer abZ4’s concerns. I will reserve my final recommendation until the empirical fairness and the robustness of the reported SOTA results are fully verified.

---

> > > ### Author Response · Authors · 2026-03-03
> > >
> > > Thank you for the follow-up. The manuscript explicitly states that all baselines were trained using their official codebases and recommended hyperparameters, and that the full baseline configurations are reported in Table 8 (Appendix 8.2). This is stated in Section 5.3 and Appendix 8.2.
> > > Table 8 further specifies that the key hyperparameters (including epochs) were copied directly from the original papers; for many baselines, the reported setting is indeed 10 epochs (e.g., DLinear, TimesNet, FiLM, iTransformer, TSLANet).
> > > We hope this addresses the concern regarding the training budget: our evaluation protocol was designed to remain consistent with prior benchmark methodology and with the settings reported as optimal in the original baseline works.

---

> > > > ### Comment · Reviewer_tFZ4 · 2026-03-03
> > > > **Concern Regarding Baseline Convergence and Public Discussion**
> > > >
> > > > I acknowledge the authors' clarification regarding the source of the baseline hyperparameters.
> > > >
> > > > However, simply replicating hyperparameter settings from original papers (e.g., the 10-epoch limit) does not automatically guarantee a fair comparison in a new experimental framework. It is the authors' responsibility to ensure that every baseline is sufficiently trained to full convergence within the current pipeline. Without verifying that the baseline performance has plateaued, the claimed superiority of NFT remains empirically questionable.
> > > >
> > > >  I strongly encourage the authors to engage in a direct and public discussion with Reviewer abZ4 regarding these convergence concerns. A robust defense of the model's state-of-the-art (SOTA) performance requires addressing the experimental fairness issues raised by all reviewers. I will refrain from making a final recommendation until the convergence of the baselines is empirically demonstrated and the concerns raised by Reviewer abZ4 are adequately resolved in the public forum.

---

> ### Author Response · Authors · 2026-03-05
>
> Thank you for the follow-up and for emphasizing the importance of this point. We agree that reusing baseline hyperparameters from the original papers (including a 10-epoch budget) does not, by itself, guarantee fairness under a different pipeline.
>
> To address this concern, we added an explicit convergence analysis in the revised manuscript: **Figure 7** and **Table 9** in **Appendix Sec. 8.2.2**, which show training and validation MSE curves (under our pipeline) for representative baselines on **ETTm1 (horizon 192)**. The purpose of this analysis is to directly inspect whether the original training budget led to obvious under-training. In this representative setting, the validation curves approach a plateau (or improve only marginally) before epoch 10, while training loss may continue to decrease slightly. Consistent with this behavior, when early stopping is enabled based on validation MSE (as in the original code), training terminates after fewer than 10 epochs.
>
> We therefore did not observe evidence of strong sensitivity to the exact training iteration count in this inspected setting, and we added this analysis to make the convergence behavior explicit in the manuscript.
>
> Regarding Reviewer abZ4’s comments, we have addressed the convergence concern in the public discussion and added the corresponding convergence plots to the paper. At the time of writing, Reviewer abZ4 has not posted additional follow-up questions, but we are happy to clarify further if needed.

---

### Decision · Action_Editor_n4XS · 2026-04-24

**Recommendation:** Accept as is

**Audience:**

Yes

**Audience Explanation:**

Yes. The paper is likely to be of interest to a meaningful portion of the TMLR audience, especially researchers working on time series forecasting, frequency-domain learning, neural signal processing, and efficient alternatives to Transformer-based sequence models. The main idea of modeling multivariate forecasting through learned multi-dimensional Fourier coefficients provides an interesting perspective on how temporal and cross-variable structure can be represented in the frequency domain. Even though some aspects of the variate-dimension interpretation require careful framing, the empirical results and subsequent analyses suggest that this modeling choice can be practically effective. The paper may also be useful to applied forecasting researchers because it provides extensive comparisons, ablation studies, and additional discussion of when the proposed method is most beneficial. Its findings could stimulate further work on frequency-domain architectures, permutation-robust spectral representations, and specialized models for strongly seasonal multivariate time series.

**Claims And Evidence:**

Yes

**Claims Explanation:**

Overall, the main claims of the submission are supported by sufficiently accurate and convincing evidence. The paper proposes Neural Fourier Transform for multivariate time series forecasting and evaluates it on multiple benchmark datasets against a broad set of baselines, with results showing consistent gains, especially on datasets with strong seasonal or periodic structure. During the discussion, the authors addressed several important concerns raised by the reviewers, including the interpretation of applying DFT along the variate dimension, the robustness to variable ordering, the role of the linear/global mixing alternatives, the sensitivity to Fourier granularity, the contribution of the trend block, and the comparison with a time-series foundation model. They also provided additional details on data splits, baseline configurations, confidence intervals over more random seeds, and a convergence analysis for representative baselines. Some reservations remain: the empirical advantage appears less pronounced on weakly seasonal data, and the comparison protocol still depends on specific dataset sources and training configurations. Nevertheless, after revision and discussion, the evidence is generally adequate to support the central claim that the proposed Fourier-based architecture is an effective forecasting model, particularly for multivariate time series with clear periodic structure.